# EAPO: Expert-guided Adaptive Preference Optimization for Recommendation

## Abstract

LLM-based recommendation systems have been widely explored due to their extensive world knowledge and powerful reasoning capabilities. However, current approaches fail to fully leverage preference data to optimize for the task, which impedes the performance of LLM-based recommendations. Although Direct Preference Optimization (DPO) has achieved significant success in aligning LLMs with human preferences, its mechanism of treating all rejected items as a homogeneous group fails to effectively capture the users' diverse preferences, resulting in poor performance on fine-grained preference discrimination. Our empirical analysis reveals that nearly half of prediction errors stem from the model's inability to accurately distinguish between chosen items and high-preference rejected items with subtle differences. To address this challenge, we propose an expert-guided adaptive preference optimization (EAPO) framework that pre-trains a lightweight recommendation model as an expert to assign personalized weights to preference sample pairs. Based on theoretical analysis, we design an adaptive $\beta$ strategy: applying smaller $\beta$ values to item pairs with similar preference levels to amplify reward differences, while using larger $\beta$ values for item pairs with significant preference disparities to ensure learning stability. Experimental results demonstrate that EAPO not only achieves superior performance in multiple benchmark datasets, but also demonstrates plug-and-play compatibility with a variety of existing preference optimization methods, establishing a new and scalable paradigm in this field.

## 1 Introduction

Recommendation systems have become crucial components in real-world applications, widely deployed across domains such as e-commerce and social media(Fang et al., 2020; Hou et al., 2023). Despite significant advancements in recommendation technologies over the past decades, modern systems still face fundamental limitations—particularly in their ability to understand users' underlying motivations and preferences. This limitation is especially pronounced in complex scenarios where user intent is implicit or expressed through natural languageAdomavicius2005,Koren2009. The emergence of Large Language Models (LLMs) provides a new opportunity to address this challenge(Wu et al., 2024b; Liu et al., 2023b).

However, there exists a significant mismatch between the training objectives of current LLM-based recommendation algorithms and the goals of personalized ranking tasks(Xu et al., 2024b; Rendle, 2022). Most existing approaches employ language modeling loss (i.e., autoregressive next-token prediction) to implement recommendation functionality(Bao et al., 2023; Liao et al., 2023; Geng et al., 2024). Such methods lack specific ranking optimization mechanisms, failing to effectively differentiate users' preference intensities across various items, thereby limiting the precision of personalized recommendations. This represents a fundamental divergence from the core objective of recommendation systems—modeling users' diverse preferences. Direct Preference Optimization (DPO)(Rafailov et al., 2023) has demonstrated significant advantages in aligning LLMs with human preferences by introducing positive-negative sample comparisons and directly optimizing implicit reward models(Rafailov et al., 2023; Wu et al., 2024a; Zhang et al., 2024), enabling models to learn more precise preference differentiation capabilities. Specifically in the recommendation domain, Softmax-DPO (S-DPO)(Chen et al., 2024) enhances models' ability to distinguish preference levels across different items through the construction of multi-negative sample contrastive mechanisms and corresponding loss functions.

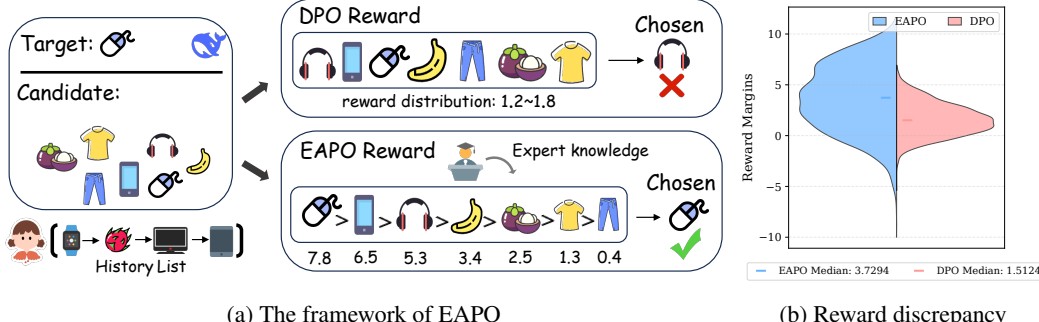

(a) The framework of EAPO (b) Reward discrepancy

Figure 1: **(a)**: The framework of EAPO. Different from existing DPO methods that treat different rejected items homogeneously, EAPO injects domain preference knowledge to help the model understand more fine-grained preference relationships among users. **(b)**: Distribution of reward discrepancy after fine-tuning with EAPO and DPO algorithms.

While traditional DPO methods optimize reward models to ensure chosen items receive higher reward than rejected items, this approach treats all rejected items as a homogeneous group, neglecting the fine-grained hierarchy of preference differences between items. *In practical recommendation scenarios, users' preference distributions across items manifest complex, multi-level structures rather than simple binary relationships.* This complexity parallels challenges in image recognition, where distinguishing similar categories (e.g., roses from carnations) requires more nuanced representations than differentiating markedly distinct categories (e.g., roses from cats). As illustrated in Figure 1b, the reward discrepancy distribution in standard DPO predominantly concentrates within a limited range, indicating its constrained ability to differentiate preference degrees across various samples and inadequate capacity to fully capture users' nuanced preferences among rejected items. Our empirical analysis further quantifies this limitation: results demonstrate that over 40% of model prediction errors originate from its inability to accurately distinguish between chosen items and rejected items with relatively high preference levels, highlighting the inherent deficiencies of existing methods in handling subtle preference variations. Notably, the parameter $\beta$ in DPO governs the model's sensitivity to reward differences(Wu et al., 2024a), yet traditional approaches employ a fixed $\beta$ value, unable to dynamically adjust according to preference judgment tasks of varying difficulty, further constraining the model's capacity to capture fine-grained preference nuances.

Based on these insights, we propose an **Expert-Guided Adaptive Preference Optimization(EAPO) framework**, a theoretically-grounded and instance-level optimization paradigm. The core innovation of this framework lies in integrating the distinctive strengths of recommendation domain expert models—their ability to accurately capture preference relationships between items and user preference patterns—into the preference learning process. Specifically, we pre-train a lightweight recommendation model as an expert model to assign personalized preference weights to each preference sample pair. Leveraging these preference weights, we design a theoretically-driven adaptive $\beta$ strategy: (1) For item pairs identified by the expert model as having similar preference levels (i.e., samples with relatively small reward differences), we apply smaller $\beta$ values. Theoretical analysis (see Section 3.1) demonstrates that this approach amplifies reward differences, enabling the model to discern these subtle yet critical preference distinctions. (2) For item pairs with larger preference disparities (i.e., samples with greater reward differences), we employ higher $\beta$ values to ensure more stable learning dynamics. Moreover, this adaptive strategy is a plug-and-play module that can be seamlessly integrated into various preference optimization methods, such as IPO(Azar et al., 2023), CPO(Xu et al., 2024a), and S-DPO(Chen et al., 2024), yielding substantial performance gains.

**The main contributions of this paper are as follows:** (i) We propose a novel optimization paradigm named EAPO, which enhances the model's ability to distinguish fine-grained preferences by combining domain expert knowledge with theoretically motivated adaptive optimization. (ii) We theoretically establish a non-monotonic relationship between the hyperparameter $\beta$ and the growth of reward margin, and derive a critical point that determines the optimization direction, thus providing a mathematical foundation for the strategy's effectiveness. (iii) Extensive experiments on multiple benchmark datasets not only validate the superior performance of our method but also highlight its significant potential as a plug-and-play module, opening new avenues for research in preference optimization.

## 2 PRELIMINARY

**Task Formulation.** We formalize the recommendation task as a language modeling problem. Let $U$ denote the user space and $I$ denote the item space. A LLM-based recommendation system $M_\theta$ receives as input a prompt containing a user $u \in U$'s interaction history and a set of candidate items $C = \{i_j\}_{j=1}^N$, and generates a response $i_p \in C$ such that $i_p$ is the candidate item that best matches user $u$'s preferences.

**Supervised Fine-Tuning of LLM4Rec.** Existing LLM-based recommendation systems primarily adapt pre-trained language models (PLMs) to perform recommendation tasks through Supervised Fine-Tuning (SFT)(Ouyang et al., 2022). This process first converts user-item interaction data into text pairs $(h_u, y_p)$, where $h_u$ represents the prompt containing the user's historical interactions and candidate items, while $y_p$ denotes the textual representation of the target item $i_p$. Subsequently, the language model is fine-tuned by maximizing the log-likelihood of the conditional generation probability, modeling the recommendation task as predicting the next token in a sequence based on context.

**Direct Preference Optimization.** Recently, Direct Preference Optimization (DPO)(Rafailov et al., 2023) has made significant progress in enhancing LLMs' ability to model human preferences. Given a dataset of triplets containing user prompts, preferred answers, and non-preferred answers$D_{pd} = \{(h_u, y_p, y_d)\}$, DPO optimizes the following objective function:

$$\mathcal{L}_{DPO} = -\mathbb{E}_{(h_u, y_p, y_d)} \left[ \log \sigma \left( \beta \log \frac{\pi_\theta(y_p|h_u)}{\pi_{ref}(y_p|h_u)} - \beta \log \frac{\pi_\theta(y_d|h_u)}{\pi_{ref}(y_d|h_u)} \right) \right] \quad (1)$$

where $\pi_\theta$ represents the policy to be optimized, $\pi_{ref}$ is the reference policy (typically a supervised fine-tuned model), $\sigma$ is the sigmoid function, and $\beta$ is a hyperparameter that regulates the trade-off between preference optimization strength and deviation from the reference policy. Rafailov et al.(Rafailov et al., 2023) demonstrates that its optimization objective is equivalent to maximizing the reward discrepancy between preferred and non-preferred items $r(h_u, y_p) - r(h_u, y_d)$, where the reward function can be implicitly expressed as:

$$r(h_u, y) = \beta \log \frac{\pi_\theta(y|h_u)}{\pi_{ref}(y|h_u)} \quad (2)$$

By minimizing $\mathcal{L}_{DPO}$, the model learns to increase the reward values for preferred items while decreasing reward values for non-preferred items, thereby enhancing its ability to model user preferences. However, standard DPO only considers binary preference relationships, ignoring fine-grained differences in preference intensity, which limits the model's ability to capture complex user preference structures, particularly in recommendation systems where users' preferences for different items typically exhibit multi-level differences.

## 3 METHODOLOGY

### 3.1 THEORETICAL MOTIVATION FOR ADAPTIVE OPTIMIZATION

Preference sample pairs $(h_u, y_p, y_d)$ in recommendation systems exhibit varying degrees of preference differentiation, necessitating a differentiated processing strategy: when selected and rejected items have similar preference levels (such as content of the same type with only slight rating differences), the model requires stronger discriminative signals to amplify reward discrepancy, thereby enhancing preference differentiation capability; conversely, when preference differences are significant (such as specific content types never encountered by users), the model can already effectively learn these differences, and in this case, stable reward discrepancy updates should be ensured to prevent overfitting.

As shown in Eq.1, the standard DPO method adopts a fixed hyperparameter $\beta$, which exhibits obvious limitations when processing multi-level preference differences. The parameter $\beta$ plays a crucial role in preference optimization, controlling the model's sensitivity and learning intensity toward preference data(Wu et al., 2024a). Therefore, for datasets with multi-level intrinsic preference

differences, the optimal $\beta$ value should be dynamically adjusted based on the characteristics of each preference sample pair, rather than using a globally fixed value.

To construct a reasonable $\beta$ dynamic adaptation mechanism that satisfies the growth trend requirements of reward discrepancies for different preference pairs, we first conduct a gradient analysis of the DPO loss function (complete derivation in Appendix A.1), obtaining:

$$\nabla_\theta \mathcal{L}_{\text{DPO}}(\theta) = -\mathbb{E}_{(h,y_p,y_d)} \left[ \frac{\beta\delta}{1 + e^{\beta\Delta r}} \right] \qquad (3)$$

where $\theta$ represents the model parameters, $\delta = \nabla_\theta r_\theta(h, y_p) - \nabla_\theta r_\theta(h, y_d)$ denotes the reward gradient difference between preferred and non-preferred samples, and $\Delta r = r_\theta(h, y_p) - r_\theta(h, y_d)$ represents the reward discrepancy.

Through theoretical analysis of the model parameter update process (detailed in Appendix A.2), we prove that during the update process from parameter $\theta_t$ to $\theta_{t+1}$, the growth amount $\Delta(\Delta r)$ of reward discrepancy $\Delta r$ is positively correlated with the following expression:

$$F(\Delta r) = \mathbb{E}_{(h,y_p,y_d)} \left[ \frac{\beta||\delta||^2}{1 + e^{\beta\Delta r}} \right] \qquad (4)$$

where $||\delta||^2$ is a positive constant. For ease of analysis, we define the gradient factor $G(\beta, \Delta r)$ as:

$$G(\beta, \Delta r) = \frac{\beta}{1 + e^{\beta\Delta r}} \qquad (5)$$

This analysis reveals that during model updates, the growth dynamics of reward discrepancy $\Delta r$ exhibit a clear positive correlation with the gradient factor $G(\beta, \Delta r)$. Specifically, $G(\beta, \Delta r)$ serves as a deterministic parameter that quantifies the model's sensitivity to preference differences during the learning process, thereby regulating the evolution of reward discrepancies.

To analyze the impact of $\beta$ on the gradient factor $G$, we compute the partial derivative of $G$ with respect to $\beta$:

$$\frac{\partial G}{\partial \beta} = \frac{1 + e^{\beta\Delta r}(1 - \beta\Delta r)}{(1 + e^{\beta\Delta r})^2} \qquad (6)$$

Through sign analysis of this partial derivative (complete proof in Appendix A.3), we demonstrate the existence of a unique critical point $\beta_c = \frac{z_c}{\Delta r}$, where $z_c$ is the numerical solution to the equation $e^{z_c}(z_c - 1) = 1$ (approximately 1.278). This critical point divides the characteristic influence of $\beta$ into two regions:

- When $\beta < \beta_c$, $\frac{\partial G}{\partial \beta} > 0$: Increasing $\beta$ accelerates the growth of gradient factor $G$, thereby accelerating the growth of $\Delta r$.

- When $\beta > \beta_c$, $\frac{\partial G}{\partial \beta} < 0$: Increasing $\beta$ leads to a decrease in gradient factor $G$, thereby slowing down the growth of $\Delta r$.

This non-monotonic property indicates that $\beta$ plays a dual role in the evolution of reward discrepancies, functioning as both a gain regulator and an inhibitory factor, depending on its specific value relative to the critical threshold $\beta_c$.

Experimental results show that the reward discrepancy $\Delta r$ between samples in the late training stages typically stabilizes above a constant value $\gamma$, making the critical value:

$$\beta_c = \frac{z_c}{\Delta r} < \frac{z_c}{\gamma}. \qquad (7)$$

At this point, if the designed adaptive strategy ensures that $\beta_{min} > \frac{z_c}{\gamma} > \beta_c$, then the growth rate of reward discrepancy $\Delta r$ can be guaranteed to decrease as $\beta$ increases.

Inspired by the above theoretical analysis, we propose a strategy to adaptively adjust the $\beta$ parameter during training: for sample pairs with large preference differences, adopting a larger $\beta$ value can reduce the growth rate of reward discrepancies, implementing a conservative learning strategy to prevent over-amplification of existing preference advantages; for sample pairs with relatively small preference differences, adopting a smaller $\beta$ value can accelerate the growth of reward discrepancies, thereby more effectively capturing these subtle but critical preference distinctions.

The practical application of this theoretical framework faces a core technical challenge: how to objectively quantify the multi-level preference differences in preference pairs, and accordingly construct a $\beta$ adjustment strategy that conforms to theoretical analysis. To address this challenge, we propose introducing pre-trained specialized recommendation models as an objective measurement benchmark for preference strength assessment, to achieve precise control over the preference learning process.

### 3.2 Expert-guided Adaptive Preference Optimization

We propose utilizing a pre-trained lightweight recommendation model as a domain expert module to precisely capture collaborative filtering relationships between items and multi-level preference structures. Despite Large Language Models (LLMs) demonstrating excellence in semantic understanding and general reasoning, traditional recommendation systems maintain significant advantages in processing user-item interaction data—efficiently leveraging statistical patterns from historical interactions and identifying fine-grained preference associations between items through collaborative filtering mechanisms, which is crucial for achieving precise preference strength assessment.

To construct an effective domain expert model, we transform content features into unique ID representations and employ the SASRec(Kang & McAuley, 2018) architecture as the backbone network for building a user behavior sequence model, an architecture that has proven its efficacy in sequential recommendation tasks. During the pre-training phase, we follow SASRec's training strategy, optimizing the model by maximizing the conditional probability of the next item in the sequence, enabling the model to master feature distributions across different items and consequently capture collaborative relationships and co-occurrence patterns between items, forming an intrinsic understanding of the dynamic evolution of user preferences. After pre-training, we freeze the model parameters and utilize it as a specialized preference strength evaluator. Formally, we define the evaluator as $f(x_u, C) \rightarrow \mathbf{S} \in \mathbb{R}^{|C|}$, where $x_u \in \mathbb{R}^{n \times d}$ represents the embedding representation of a user's historical sequence of length $n$, $C$ denotes the candidate set composed of chosen and rejected items, and $d$ is the feature embedding dimension. This function outputs a score vector $\mathbf{S} = [s_1, s_2, ..., s_{|C|}]$, where $s_i$ indicates the user's preference level for candidate item $c_i$, reflecting the relative preference strength and collaborative correlations between candidate items.

Building upon this expert scoring mechanism, we further design a method to precisely quantify preference differences. Specifically, for calculating the weight coefficient of a preference sample $(h_u, y_p, y_d)$, we first input the user's historical ID sequence $x_u$ into the expert model to obtain prediction scores for all items in the candidate set: $\mathbf{S} = f(x_u, C)$. Subsequently, we extract the score of the chosen item $S_{y_p}$ and the scores of each item $S_{y_d}$ in the rejected item set $C_l$, computing the preference difference degree between the chosen item and each rejected item:

$$w_{(y_p, y_d)} = S_{y_p} - S_{y_d}, \quad d \in C_l \tag{8}$$

where a larger $w_{(y_p, y_d)}$ indicates a greater degree of preference difference between the two. Based on this difference metric, we design an adaptive $\beta$ value calculation method:

$$\beta_{(y_p, y_d)} = \frac{w_{(y_p, y_d)} - \min_{j \in C_l} w_{(y_p, y_j)}}{\max_{j \in C_l} w_{(y_p, y_j)} - \min_{j \in C_l} w_{(y_p, y_j)}} \times (U_\beta - \frac{z_c}{\gamma}) + \frac{z_c}{\gamma} \tag{9}$$

where $U_\beta$ is the upper bound of the normalization interval. This normalized coefficient design has important theoretical significance: according to the analysis in Section 3.1, when $\Delta r > \gamma$ during the training process, if $\beta \geq \frac{z_c}{\gamma}$, it can guarantee that the growth rate of the reward discrepancy $\Delta r$ decreases as $\beta$ increases. Our method ensures that $\min \beta_{(y_p, y_d)} \geq \frac{z_c}{\gamma}$, therefore when the preference levels between the chosen and rejected items are highly similar, the $\beta_{(y_p, y_d)}$ value is smaller, contributing to a larger growth magnitude of $\Delta r$, enabling the model to more effectively capture these subtle preference distinctions; when the preference difference between the two is significant, the $\beta_{(y_p, y_d)}$ value is larger, achieving conservative model updates and preventing overfitting.

Through the above mechanism, each rejected item receives a weight value within the interval $[\frac{z_c}{\gamma}, U_\beta]$ based on its preference proximity to the chosen item. This normalization framework not only precisely quantifies the relative preference difference between each rejected item and the chosen item but also fairly reflects the preference hierarchy structure among different rejected items, providing more fine-grained signals for subsequent preference optimization.

Finally, we integrate the adaptive $\beta$ strategy into the Direct Preference Optimization (DPO) framework, redefining the loss function as:

$$\mathcal{L}_{\text{EAPO}} = -\mathbb{E}_{(h_u, y_p, y_d) \sim \mathcal{D}} \left[ \log \sigma \left( \beta_{(y_p, y_d)} \cdot (r_\theta(h_u, y_p) - r_\theta(h_u, y_d)) \right) \right] \quad (10)$$

This expert model-guided adaptive preference optimization framework effectively combines the preference perception capability of traditional recommendation systems with the semantic understanding advantages of LLMs, providing a more refined gradient control mechanism for preference optimization. Notably, the expert model is exclusively utilized during the training phase to compute preference weights and is decoupled from the inference process. Therefore, EAPO imposes no additional computational overhead on the recommendation system at inference stage.

### 3.3 GENERALITY ANALYSIS

We observe that the adaptive $\beta$ strategy demonstrates universal applicability in the field of preference learning. This dynamic regulation mechanism seamlessly integrates with mainstream preference optimization methods (such as IPO, CPO, S-DPO, etc.), highlighting its value as a methodological foundation innovation. Based on these observations, we propose a unified adaptive framework:

$$\mathcal{L}_{\text{AP}}(\theta) = f(\beta_{(y_p, y_d)}, r_\theta(h_u, y_p), r_\theta(h_u, y_d)) \quad (11)$$

where $f$ represents the loss function of a specific preference learning method, and $\beta_{(y_p, y_d)}$ is the regulation coefficient dynamically calculated based on the reference model's evaluation.

Taking the IPO algorithm based on square loss function as an example, its adaptive form can be expressed as (see Appendix C.5 for integration with other algorithms):

$$\mathcal{L}_{\text{A-IPO}}(\theta) = \left( \beta_{(y_p, y_d)}(r_\theta(x_u, y_p) - r_\theta(x_u, y_d)) - \frac{1}{2\tau} \right)^2 \quad (12)$$

The adaptive $\beta$ strategy reveals a key insight into preference learning: when preference pairs exhibit varying levels of discrimination difficulty, preference learning algorithms should receive differentiated optimization signals. This finding facilitates a paradigm shift from "static uniform optimization" to "dynamic adaptive optimization." Importantly, our method does not rely on specific preference probability models (such as the Bradley-Terry model(Bradley & Terry, 1952)) but provides a universal enhancement mechanism. Regardless of the mathematical formulation adopted by the underlying preference modeling, the dynamic $\beta$ regulation mechanism can effectively enhance the learning process, allowing non-BT model methods, including IPO, to benefit significantly. Experimental results (see Section 4.4)) validate the effectiveness of this method across various preference learning frameworks.

## 4 EXPERIMENTS

In this section, we aim to address the following research questions:

- **RQ1:** How does EAPO perform compared to with traditional and LLMs-based recommenders on performance?
- **RQ2:** How do LLMs-based recommenders benefit from $\beta$ in adaptive preference optimization?
- **RQ3:** How does the quality of the expert model affect the performance of EAPO?
- **RQ4:** How generalizable is the EAPO with other preference optimization algorithms?

### 4.1 EXPERIMENTAL SETUP

**Baseline Models.** We conducted comprehensive comparisons between EAPO and three categories of sequential recommendation systems: (1) traditional recommender systems, including GRU4Rec(Hidasi et al., 2016) Caser(Tang & Wang, 2018), and SASRec(Kang & McAuley, 2018); (2) language model-based recommendation systems, including LLaMA3-8B(Dubey et al., 2024), Qwen2.5-7B(Yang et al., 2025), Chat-REC(Gao et al., 2023), TALLRec(Bao et al., 2023), and LLaRA (Liao et al., 2023); and (3) preference optimization-based recommendation systems(S-DPOChen et al.

Table 1: The performance comparison on three industrial datasets.

| | | Movies and TV | | | Books | | | Pet Supplies | | |
|---|---|---|---|---|---|---|---|---|---|---|
| | | HR@1 | HR@5 | NDCG@5 | HR@1 | HR@5 | NDCG@5 | HR@1 | HR@5 | NDCG@5 |
| Traditional | GRU4Rec | 0.1289 | 0.2142 | 0.1728 | 0.0391 | 0.1187 | 0.0829 | 0.1487 | 0.1165 | 0.0793 |
| | Caser | 0.0264 | 0.1304 | 0.0406 | 0.1347 | 0.3148 | 0.1069 | 0.0323 | 0.1394 | 0.0423 |
| | SASRec | 0.1847 | 0.3783 | 0.2842 | 0.2537 | 0.5160 | 0.3914 | 0.1789 | 0.3583 | 0.2713 |
| LM-based | LLaMA3 | 0.0581 | 0.1795 | 0.1326 | 0.0489 | 0.1349 | 0.0949 | 0.0441 | 0.0966 | 0.0675 |
| | Qwen 2.5 | 0.0349 | 0.1147 | 0.0862 | 0.0281 | 0.1178 | 0.0914 | 0.0212 | 0.1464 | 0.1062 |
| | ChatRec | 0.2271 | 0.4419 | 0.3056 | 0.1702 | 0.3151 | 0.2252 | 0.2084 | 0.3899 | 0.2874 |
| | TALLRec | 0.1037 | 0.2643 | 0.2726 | 0.3827 | 0.5866 | 0.5322 | 0.1097 | 0.2519 | 0.2712 |
| | LLaRA | 0.1156 | 0.2725 | 0.2133 | 0.4267 | **0.7028** | 0.5812 | 0.1223 | 0.2597 | 0.2122 |
| PO-based | S-DPO | 0.4841 | 0.6597 | 0.5781 | 0.4369 | 0.6954 | 0.5729 | 0.3179 | 0.5810 | 0.4314 |
| | **EAPO** | **0.5760** | **0.6977** | **0.6331** | **0.5356** | 0.6990 | **0.6187** | **0.3385** | **0.5838** | **0.4539** |

(2024), $\beta$-DPOWu et al. (2024a). For detailed descriptions and comparative analyses of the baseline models, please refer to AppendixB.1.

**Datasets and Implementation.** We conducted extensive experiments on three publicly available industrial datasets covering different domains: Movies and TV(Ni et al., 2019b), Books(Ni et al., 2019a), and Pet Supplies(Ni et al., 2019c). For implementation, we utilized Llama-3-8B(Dubey et al., 2024) as the backbone network and fine-tuned its parameters using LoRa during both the Supervised Fine-Tuning (SFT)(Ouyang et al., 2022) and preference alignment stages. To evaluate recommendation performance, we adopted we employed Hit Rate (HR@K) and Normalized Discounted Cumulative Gain (NDCG@K) as performance metrics, with k values of 1 and 5. For a comprehensive description of the datasets, implementation and metrics details, please refer to Appendix B.2.1.

## 4.2 Performance Comparison (RQ1)

As shown in Table 1, we conducted a comprehensive evaluation of EAPO against baseline models using the critical ranking metrics in recommendation systems—HR@1, HR@5, and NDCG@5—across three industrial datasets from different domains.

The experimental results indicate that while traditional recommenders and supervised fine-tuned (SFT) LLM recommenders each have their respective strengths, preference optimization-based methods exhibit significant performance improvements over both. This highlights the importance of considering inter-item preference differences during model optimization. Notably, EAPO consistently outperforms all baseline models, including both preference-optimized and SFT approaches. This demonstrates the critical role of our theoretically-grounded framework in quantifying multi-level preference differences among items, which in turn enhances the quality of model training.

Specifically, on the HR@1 metric, EAPO achieves substantial improvements ranging from 6.4% to 22.58% over the baselines across the three datasets. This result demonstrates that EAPO can finely distinguish multi-level preference relationships, enabling items truly preferred by the user to rank highest in the model's internal scoring, thereby boosting overall recommendation performance. Furthermore, for NDCG@5, a comprehensive metric that measures ranking quality, EAPO also surpasses all baselines, with improvements reaching up to 9.52%. This further validates that EAPO not only helps the model capture the distinction between preferred and rejected items but also enables it to understand the relative preference relationships among different rejected items. Consequently, it demonstrates a more pronounced advantage on position-aware metrics like NDCG.

## 4.3 Ablation Study

To investigate the impact and robustness of the adaptive preference $\beta$, we conducted the analysis from three key perspectives: (1) optimization effect on ranking evaluation metrics, (2) enhancement effect on preference margins, and (3) robustness in cold-start and cross-domain scenarios.

**Ranking Evaluation (RQ2).** As illustrated in Figure 2a, we compared EAPO with the SFT, the standard DPO algorithm, and $\beta$-DPO with heuristic dynamic $\beta$. Results show that dynamic preference methods outperform static preference tuning and SFT models, and that EAPO achieves notably better performance than $\beta$-DPO. This demonstrates that our adaptive strategy, guided by expert knowledge

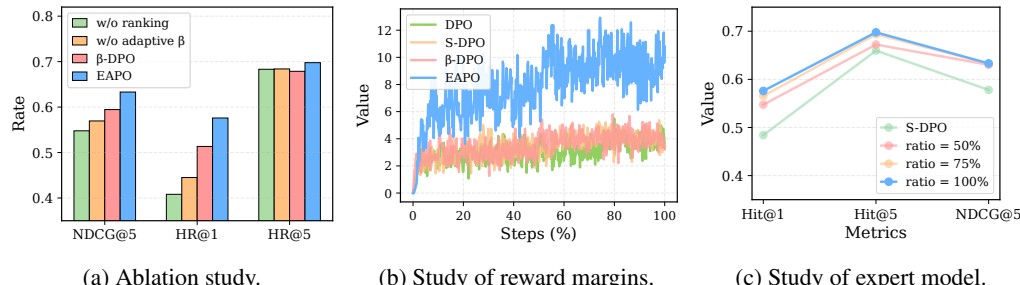

(a) Ablation study.  (b) Study of reward margins.  (c) Study of expert model.

Figure 2: Studies on values of $\beta$ of *EAPO* on Movie and TV. **(a)** Ablation study of EAPO compared with SFT, DPO and $\beta$-DPO on three metric. **(b)** Comparison of reward margins among DPO, S-DPO, $\beta$-DPO and EAPO algorithms. **(c)** Performance comparisons with different quality of expert model.

Table 2: EAPO combined with other preference algorithms (relative gain in red).

| Metric | S−DPO | EA−SDPO | CPO | EA−CPO | IPO | EA−IPO |
|--------|-------|---------|-----|--------|-----|--------|
| HR@1 | 0.4841 | **0.5558** (+14.81%) | 0.4341 | **0.4422** (+1.87%) | 0.4071 | **0.4209** (+3.39%) |
| HR@5 | 0.6597 | **0.6829** (+3.52%) | 0.6345 | **0.6628** (+4.46%) | 0.7024 | **0.7143** (+1.69%) |
| NDCG@5 | 0.5781 | **0.6118** (+5.83%) | 0.5091 | **0.5359** (+5.26%) | 0.5456 | **0.5568** (+2.05%) |

and theoretical principles, captures users' multi-level preference relationships more accurately and effectively than heuristic strategies that rely solely on intra-batch statistics.

**Preference Margin Analysis (RQ2).** To verify whether EAPO amplifies the reward gap between preferred and non-preferred samples, we compared the evolution of reward margins during training for EAPO, DPO, S-DPO and $\beta$-DPO. As shown in Figure 2b, EAPO achieves faster and larger increases in reward margins between preferred and rejected samples. This indicates that EAPO not only strengthens the advantage of preferred items but also widens the preference gaps among different rejected items, enabling the model to more precisely distinguish their degree of relevance. Appendix C.1 further analyzes reward distributions and EAPO's ability to discriminate high-rejection items with nuanced preference signals, corroborating its ability to improve recommendation accuracy through fine-grained preference alignment.

**Robustness in Challenging Settings (RQ2).** To evaluate the effectiveness and stability of EAPO in complex real-world scenarios, we conducted detailed experiments to investigate its performance under two major challenges: data sparsity (cold-start) and domain shift (cross-domain), with further details provided in Appendix C.2. These experiments not only validate the robustness of our method but also reveal the unique advantages of the expert-guided paradigm in handling such complex scenarios.

**Impact of Expert Model Quality on EAPO (RQ3).** To examine EAPO's sensitivity and robustness to expert quality, we systematically reduce the quality of the expert by downsampling the training interaction data at different ratios. Specifically, we set the ratio to 75% and 50%, respectively, and trained two weaker expert models to replace the original expert scorer. As shown in Figure2c, when training data is reduced from 100% to 50%, EAPO's HR@1 drops only slightly (by approximately 0.0286), while HR@5 and NDCG@5 remained at comparable levels. This demonstrates that even when the expert model exhibits systematic biases or degraded accuracy, as long as their judgments on relative preference rankings remain generally stable, EAPO can still effectively utilize reliable collaborative signals to guide LLM preference alignment, thereby maintaining high overall ranking performance and demonstrating enhanced transferability and robustness in real-world industrial scenarios.

### 4.4 GENERALITY ANALYSIS (RQ4)

As described in Section 3.3, our proposed adaptive preference parameter $\beta$ methodology can be directly extended to other pairwise preference optimization methods, including but not limited to IPO, CPO, and S-DPO. To validate this generalizability, we incorporated the adaptive parameter $\beta$ into the loss functions of these three methods, resulting in the corresponding EA-IPO, EA-CPO, and EA-SDPO models. Table 2 quantitatively demonstrates the performance of various algorithms across recommendation ranking metrics. The results indicate that after adaptive parameter optimization,

all methods achieved performance improvements across all metrics, with HR@1 increasing by up to 14.81%, further substantiating the broad applicability and effectiveness of our proposed adaptive preference optimization approach. In Figure 5 of the Appendix, we illustrate how introducing the adaptive parameter affects reward growth across different algorithms. Experimental results confirm that, regardless of the underlying algorithm, incorporating quantitative preference differential assessment can further enhance the preference advantage of chosen items over rejected items.

## 5 RELATED WORK

**LLM-based Recommender.** LLMs have gained significant traction in recommender systems due to their expansive knowledge base and advanced reasoning capabilities(Fan et al., 2023; Zhao et al., 2023; Wei et al., 2024). This integration has evolved along two distinct paradigms: LLM-enhanced and LLM-based recommender systems. The former(Gao et al., 2023; Lin et al., 2023; Ren et al., 2023) supplements conventional recommendation algorithms with LLM capabilities while still relying on traditional architectures, thus underutilizing the inherent reasoning potential of language models. Conversely, LLM-based systems(Bao et al., 2023; Liao et al., 2023; Geng et al., 2022) directly employ language models as the primary recommendation engine. However, unmodified LLM-based recommenders exhibit deficiencies in instruction-following and domain-specific expertise. To mitigate these limitations, researchers have increasingly focused on supervised fine-tuning of LLMs using historical interaction data(Xu et al., 2024b; Bao et al., 2023; Zhang et al., 2023b). Contemporary investigations reveal that optimizing item representation methodologies during fine-tuning can substantially enhance recommendation performance(Hua et al., 2023). These advancements encompass: incorporating collaborative signals(Liao et al., 2023; Yang et al., 2023; Li et al., 2023), optimizing numerical representations(Rajput et al., 2023), and integrating supplementary item embeddings(Geng et al., 2022; Zhu et al., 2023; Zheng et al., 2023). Nevertheless, existing fine-tuning approaches primarily adhere to language generation objectives without specifically addressing personalized preference modeling. In contrast, our proposed framework explicitly optimizes item ranking information derived from preference data, thereby directly addressing the core challenge of personalized recommendation.

**Direct Preference Optimization.** Reinforcement Learning from Human Feedback (RLHF)(Stiennon et al., 2020; Touvron et al., 2023) has emerged as a crucial methodology enabling LLMs to assimilate human preferences. The RLHF pipeline encompasses reward model training and RL optimization(Yue et al., 2023; Zhao et al., 2025; Yue et al., 2024), with the latter characterized by instability and inefficiency. Direct Preference Optimization (DPO)(Rafailov et al., 2023) circumvents the fragile RL stage through specific reward model parameterization, thereby maintaining RLHF performance while enhancing implementation accessibility. DPO has demonstrated effectiveness across multiple domains, including natural language processing(Amini et al., 2024; Zhang et al., 2023a; Dai et al., 2023) and multimodal language model applications(Zhang et al., 2024; Amini et al., 2024; Zhang et al., 2023a). Furthermore, several variants have been proposed to further refine DPO. $\Psi$PO(Azar et al., 2023) generalizes the DPO loss function, with its instantiation IPO exhibiting superior resistance to overfitting. $\beta$-DPO(Wu et al., 2024a) investigates how the hyperparameter $\beta$ influences samples of varying similarity during model updates. S-DPO(Chen et al., 2024) introduces multiple negative samples, breaking the limitation of focusing only on positive samples. However, few studies have explored DPO's performance in contexts involving multifaceted user preferences.

## 6 CONCLUSION

In this paper, we present the EAPO, addressing the critical challenge of preference representation in LLM-based recommendation systems. By introducing a domain expert model to quantify preference differences between samples and designing a theoretically-grounded adaptive $\beta$ strategy, our approach successfully enhances the model's ability to distinguish multi-level preference structures, particularly showing significant improvements when handling fine-grained preference distinctions. Experimental results validate the effectiveness of our framework across multiple benchmark datasets and demonstrate its strong compatibility with existing preference optimization methods. This work not only provides a novel perspective for addressing preference learning problems in LLMs-based recommendations, but also establishes a new paradigm for integrating LLMs with domain-specific knowledge.

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

# Appendix — Table of Contents

## A  MATHEMATICAL DERIVATIONS

### A.1  DERIVING THE GRADIENT OF EAPO LOSS

In reinforcement learning from human feedback (RLHF), a fundamental challenge lies in effectively incorporating preference data into model training. The DPO loss function provides an elegant mathematical framework for this purpose, enabling models to learn from paired preference examples.

DPO loss function is defined as:

$$\mathcal{L}(\theta) = -\mathbb{E}_{(h,y_p,y_d)}\Big[\log \sigma\big(\beta\big(r_\theta(h, y_p) - r_\theta(h, y_d)\big)\big)\Big] \tag{13}$$

In this formulation, $\sigma$ represents the sigmoid function which maps the score difference to a probability scale, $\beta$ serves as a temperature parameter controlling the sharpness of the preference boundary, and $r_\theta(h, y)$ denotes the model's scoring function for input $h$ and output $y$. For notational convenience, we define the score difference between preferred and non-preferred outputs as $\Delta r = r_\theta(h, y_p) - r_\theta(h, y_d)$.

To optimize this loss function through gradient-based methods, we must first compute its gradient with respect to the model parameters $\theta$:

$$\nabla_\theta \mathcal{L}(\theta) = -\mathbb{E}_{(h,y_p,y_d)}\big[\nabla_\theta \log \sigma(\beta\Delta r)\big] \tag{14}$$

This gradient calculation requires careful application of the chain rule. Let's proceed step by step to derive an explicit form:

$$\nabla_\theta \log \sigma(\beta\Delta r) = \frac{1}{\sigma(\beta\Delta r)}\nabla_\theta \sigma(\beta\Delta r) \tag{15}$$

Utilizing the well-known property of the sigmoid function's derivative, where $\sigma'(z) = \sigma(z)(1-\sigma(z))$, we can further expand:

$$\nabla_\theta \sigma(\beta\Delta r) = \sigma'(\beta\Delta r)\nabla_\theta(\beta\Delta r) = \sigma(\beta\Delta r)\big(1 - \sigma(\beta\Delta r)\big)\beta\nabla_\theta\Delta r \tag{16}$$

Substituting this result back into our earlier expression yields:

$$\nabla_\theta \log \sigma(\beta\Delta r) = \frac{\sigma(\beta\Delta r)\big(1 - \sigma(\beta\Delta r)\big)\beta\nabla_\theta\Delta r}{\sigma(\beta\Delta r)} = \beta\big(1 - \sigma(\beta\Delta r)\big)\nabla_\theta\Delta r \tag{17}$$

Through algebraic manipulation and using the identity $1 - \sigma(\beta\Delta r) = \dfrac{1}{1 + e^{\beta\Delta r}}$, we obtain a more compact form:

$$\nabla_\theta \log \sigma(\beta\Delta r) = \frac{\beta}{1 + e^{\beta\Delta r}}\nabla_\theta\Delta r \tag{18}$$

For the gradient of the preference difference, we have:

$$\nabla_\theta\Delta r = \nabla_\theta\big(r_\theta(h, y_p) - r_\theta(h, y_d)\big) = \nabla_\theta r_\theta(h, y_p) - \nabla_\theta r_\theta(h, y_d) \tag{19}$$

Introducing $\delta = \nabla_\theta r_\theta(h, y_p) - \nabla_\theta r_\theta(h, y_d)$ as a more concise notation for this gradient difference, our final expression for the loss gradient becomes:

$$\nabla_\theta \mathcal{L}(\theta) = -\mathbb{E}_{(h,y_p,y_d)}\left[\frac{\beta\,\delta}{1 + e^{\beta\Delta r}}\right] \tag{20}$$

This elegant formulation reveals how the gradient is modulated by both the temperature parameter $\beta$ and the current score difference $\Delta r$, providing crucial insights into the learning dynamics of preference optimization.

## A.2 MODEL PARAMETER UPDATE PROCESS

Having derived the gradient of the EAPO loss, we now turn our attention to understanding how the temperature parameter $\beta$ influences the parameter update process. This analysis is essential for developing optimal training strategies that balance learning speed and stability.

We begin by defining the gradient factor $G(\beta, \Delta r) = \dfrac{\beta}{1 + e^{\beta \Delta r}}$, which appears in our gradient expression and modulates the strength of parameter updates. To understand how this factor varies with $\beta$, we compute its partial derivative:

$$\frac{\partial G}{\partial \beta} = \frac{\partial}{\partial \beta}\left(\frac{\beta}{1 + e^{\beta \Delta r}}\right) = \frac{1}{1 + e^{\beta \Delta r}} - \frac{\beta \Delta r\, e^{\beta \Delta r}}{(1 + e^{\beta \Delta r})^2} \tag{21}$$

Through algebraic simplification, we obtain:

$$\frac{\partial G}{\partial \beta} = \frac{1 + e^{\beta \Delta r}\bigl(1 - \beta \Delta r\bigr)}{(1 + e^{\beta \Delta r})^2} \tag{22}$$

To identify critical points in this function, we introduce the substitution $z = \beta \Delta r$ and define $f(z) = 1 + e^z(1 - z)$. The derivative of this function is:

$$f'(z) = \frac{d}{dz}\bigl[1 + e^z(1 - z)\bigr] = e^z(1 - z) - e^z = -z\, e^z \tag{23}$$

Setting $f(z_c) = 0$ leads to the equation:

$$e^{z_c}(z_c - 1) = 1, \tag{24}$$

This transcendental equation yields the numerical solution $z_c \approx 1.278$, which translates to a critical value of $\beta_c = \dfrac{z_c}{\Delta r}$.

The identification of this critical point $\beta_c$ is mathematically significant as it precisely demarcates distinct optimization regimes in the preference learning dynamics. Our analysis reveals that the gradient factor $G$ plays a crucial role in determining the rate of growth for preference gap $\Delta r$:

- When $\beta < \beta_c$, we observe $\frac{\partial G}{\partial \beta} > 0$, indicating that increasing $\beta$ accelerates the growth of gradient factor $G$, thereby enhancing the rate at which $\Delta r$ expands. This represents an efficiency-gaining regime where higher temperature values yield proportionally better discrimination between preferred and dispreferred options.
- When $\beta > \beta_c$, we observe $\frac{\partial G}{\partial \beta} < 0$, indicating that increasing in $\beta$ actually cause a decrease in gradient factor $G$, consequently decelerating the growth of $\Delta r$. This regime introduces diminishing returns and potentially counterproductive effects in the optimization process.

This mathematical characterization provides theoretical guidance for adaptive temperature scheduling during training, allowing for optimal preference learning efficiency across different stages of model development.

## A.3 ANALYSIS OF PARTIAL DERIVATIVES

Having established the critical role of the temperature parameter, we now delve deeper into the dynamics of how model parameters evolve during training. This analysis provides valuable insights into the convergence properties and stability of the preference optimization process.

We begin with a first-order Taylor expansion to approximate the change in preference gap after a parameter update:

$$\Delta r(\theta_{t+1}) = \Delta r(\theta_t) + \nabla_\theta \Delta r(\theta_t) \cdot \Delta \theta + O\bigl(\|\Delta \theta\|^2\bigr) \tag{25}$$

Rearranging terms to focus on the change in preference gap:

$$\Delta r(\theta_{t+1}) - \Delta r(\theta_t) = \nabla_\theta \Delta r(\theta_t) \cdot \Delta\theta + O\big(\|\Delta\theta\|^2\big) \tag{26}$$

Since the first-order gradient plays a major role, we define:

$$\delta(\Delta r) = \nabla_\theta \Delta r(\theta_t) \cdot \Delta\theta. \tag{27}$$

In gradient descent optimization, the parameter update is given by:

$$\Delta\theta = -\eta\nabla_\theta\mathcal{L}(\theta) = \eta\,\mathbb{E}_{(h,y_p,y_d)}\left[\frac{\beta\,\delta}{1+e^{\beta\Delta r}}\right], \tag{28}$$

where $\eta$ is the learning rate. Substituting this into our expression for $\delta(\Delta r)$:

$$\delta(\Delta r) = \eta\,\nabla_\theta\Delta r(\theta_t) \cdot \mathbb{E}_{(h,y_p,y_d)}\left[\frac{\beta\,\delta}{1+e^{\beta\Delta r}}\right] \tag{29}$$

Since $\nabla_\theta\Delta r(\theta_t) = \delta$, we can rewrite this as an inner product:

$$\delta(\Delta r) = \eta\left\langle \delta,\ \mathbb{E}_{(h,y_p,y_d)}\left[\frac{\beta\,\delta}{1+e^{\beta\Delta r}}\right]\right\rangle \tag{30}$$

Under the assumption of unbiased gradient estimation, this simplifies to:

$$\delta(\Delta r) = \eta\,\mathbb{E}_{(h,y_p,y_d)}\left[\frac{\beta\,\langle\delta,\delta\rangle}{1+e^{\beta\Delta r}}\right] = \eta\,\mathbb{E}_{(h,y_p,y_d)}\left[\frac{\beta\,\|\delta\|^2}{1+e^{\beta\Delta r}}\right] \tag{31}$$

This final expression illuminates the complex interplay of factors governing the evolution of the preference gap during training. Specifically, the change in $\Delta r$ is determined by three key components: the learning rate $\eta$, the squared norm of the gradient difference $\|\delta\|^2$, and the gradient factor $G(\beta, \Delta r) = \dfrac{\beta}{1+e^{\beta\Delta r}}$.

The critical value $\beta_c = \dfrac{z_c}{\Delta r}$ (where $z_c \simeq 1.278$) identified earlier plays a pivotal role in this dynamic. It separates three distinct training regimes:

- When $\beta < \beta_c$: Increasing $\beta$ accelerates the growth of the preference gap $\Delta r$.
- When $\beta = \beta_c$: The growth rate of $\Delta r$ reaches its maximum efficiency.
- When $\beta > \beta_c$: Further increases in $\beta$ decelerate the growth of $\Delta r$.

These insights provide practitioners with valuable guidance for temperature scheduling strategies in preference optimization. By adaptively adjusting $\beta$ throughout training, one can potentially achieve faster convergence and better generalization performance, balancing the exploitation of strong preferences with the exploration of more ambiguous examples.

The mathematical framework developed in this analysis not only deepens our theoretical understanding of preference-based learning but also offers practical implications for implementing more efficient and effective training procedures in reinforcement learning from human feedback systems.

## B    EXPERIMENTAL SETTINGS

### B.1    BASELINES

To comprehensively validate the effectiveness of the EAPO framework, we systematically compare it with traditional sequential recommendation models and various state-of-the-art baselines based on large language models (LLMs).

### B.1.1 Traditional Sequential Recommendation Models

- GRU4Rec(Hidasi et al., 2016): This model employs a Gated Recurrent Unit (GRU) architecture for temporal modeling of user interaction sequences, effectively capturing sequential dependencies for next-item prediction tasks.
- Caser(Tang & Wang, 2018): By applying convolutional neural networks in both horizontal and vertical dimensions, this model captures local and global high-order patterns within sequences, thereby enhancing recommendation accuracy and relevance.
- SASRec(Kang & McAuley, 2018): This model incorporates self-attention mechanisms and positional encoding, utilizing multi-head attention structures to learn complex short-term and long-term dependencies between items in sequences, effectively modeling the evolution of dynamic user interests.

### B.1.2 LLM-based Recommendation Models

- QWen 2.5(Yang et al., 2025): As a zero-shot baseline, we directly employ the QWen-2.5-7B model, generating candidate item rankings through carefully designed recommendation task prompts without any additional fine-tuning.
- LLaMA3(Dubey et al., 2024): As a zero-shot baseline, we directly employ the LLaMA3-8B model, generating candidate item rankings through carefully designed recommendation task prompts without any additional fine-tuning.
- Chat-REC(Gao et al., 2023): Following the conversational recommendation framework proposed by (Liu et al., 2023a), this method uses sequences of product titles from users' historical interactions as profile inputs, and leverages Gemini-1.5-pro(Google) as the inference engine to generate personalized recommendation responses.
- TALLRec(Bao et al., 2023): This approach first converts user interaction sequences into structured textual prompts, then performs task-adaptive fine-tuning on pre-trained large language models using domain-specific corpora to enhance contextual understanding capabilities for recommendations.
- LLaRA(Liao et al., 2023): This method innovatively integrates collaborative filtering signals from traditional recommender systems into the instruction fine-tuning process of large language models, significantly improving LLM performance on recommendation tasks.

### B.1.3 Preference Optimization-based Recommendation Models

- S-DPO(Chen et al., 2024): A direct preference optimization approach based on multiple negative samples, which applies a Softmax mechanism to calculate differentiated weights for various negative samples, constructing a cross-entropy based optimization objective. This method leverages the DPO algorithm(Rafailov et al., 2023) for efficient preference data learning, enhancing recommendation relevance while maintaining generation diversity.
- $\beta$-DPO(Wu et al., 2024a): An enhanced Direct Preference Optimization framework that addresses the limitations of using a static trade-off parameter $\beta$. It introduces a novel approach that dynamically calibrates $\beta$ based on the quality of preference data to improve the model alignment process.

### B.2 Experimental Setup and Evaluation

### B.2.1 Datasets

To thoroughly evaluate the effectiveness and generalization capabilities of EAPO, we conducted systematic experiments on three widely-used real-world e-commerce datasets: Movie and TV, Books, and Pet Supplies.

- Movie and TV: This dataset contains user ratings and reviews for movies and television programs, characterized by high sparsity and diverse user preference patterns, serving as a standard benchmark dataset for evaluating recommendation algorithms in the entertainment domain.
- Books: This is a large-scale dataset comprising user reviews of book products across various categories, featuring widely distributed user interests with pronounced long-tail distribution characteristics, suitable for validating model capabilities in handling diverse content.

- Pet Supplies: This dataset records purchase histories and reviews of pet products, exhibiting stronger domain specificity and more specialized product attribute descriptions compared to the previous two datasets, enabling assessment of model adaptability in vertical domains.

Following the experimental design of (Bao et al., 2023), we use item title texts as the primary content features across all three datasets. All interaction data are strictly ordered by timestamp and divided into training, validation, and testing sets in an 8:1:1 ratio to ensure the temporal integrity of model evaluation, effectively preventing evaluation bias due to information leakage.

### B.2.2 Implementation Details

For traditional recommender baselines, we employed the Adam optimizer with learning rates, embedding dimensions, and regularization coefficients following the protocols established in(Liao et al., 2023; Chen et al., 2024). All LLM-based methods were implemented on 4 A100 NVIDIA RTX GPUs, leveraging the widely adopted Llama-3-8B as the backbone architecture. The parameters of this backbone were fine-tuned using LoRa during both the Supervised Fine-Tuning (SFT) and preference alignment phases. In our prompts, all items were represented by their titles, which served as their textual features. Unlike some alternative approaches, our method optimizes solely for item title loss while still achieving strong recommendation performance. The specifics are as follows:

- Experimental Environment: All experiments were implemented in Python 3.9.7, using PyTorch 2.2.2 as the deep learning framework and Transformers 4.38.2 for model construction; computational resources consisted of 4 NVIDIA A100 GPU accelerator cards.

- Base Model: We selected LLaMA3-8B(Dubey et al., 2024) as the backbone language model for the EAPO framework, which achieves an optimal balance between general comprehension capabilities and computational efficiency.

- Prompt Strategy: Following the methodology of (Liao et al., 2023), we adopted a multi-template sampling strategy during both training and evaluation phases, randomly selecting different expression formats from a predefined instruction template library to enhance the model's comprehension capabilities and generalization performance for diverse natural language instructions.

- Traditional Model Optimization Parameters: For traditional sequential recommendation models, we employed the Adam optimizer with an initial learning rate of 0.001 and batch size of 256, determining the optimal L2 regularization coefficient $\lambda$ through grid search within the range 1e-3, 1e-4, 1e-5, 1e-6, 1e-7.

- LLM Training Configuration: All LLM-based baseline models were trained for 5 complete epochs with a batch size of 128; model selection was based on saving the checkpoint with the lowest validation loss.

- Learning Rate Scheduling: We implemented a dynamic learning rate scheduling strategy with initial linear warm-up to peak value (occupying 5% of total training steps), followed by cosine annealing, balancing exploration and convergence efficiency during the optimization process.

- Preference Training: This training process was conducted in two phases, beginning with supervised fine-tuning for 2 epochs to establish foundational capabilities, followed by reinforcement learning-based training on preference data for 2 additional epochs, with a batch size of 32 and a constant learning rate of 1e-5.

### B.2.3 Evaluation Metrics

Considering that large language models primarily generate textual sequences rather than explicit ranking scores typical of traditional recommender systems, we adopt the re-ranking evaluation paradigm consistent with (Liao et al., 2023) to ensure fairness and comparability in experimental assessment:

- HitRatio@1 (HR@1): For each test sequence, we randomly sample 20 items with which the user has not interacted, combining them with the actual target item to form a test set of 21 candidates. This metric measures the model's ability to accurately rank the true target item in the first position, directly reflecting the precision of recommendations.

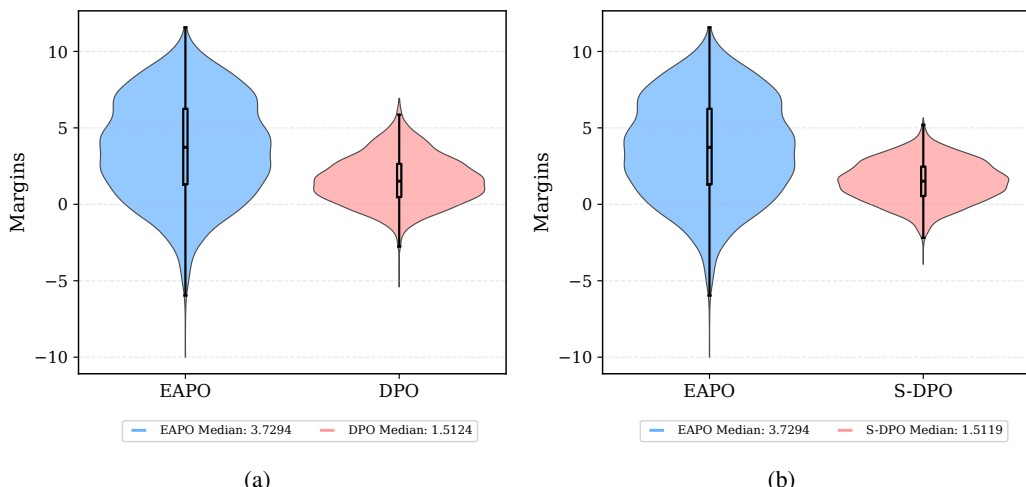

Figure 3: Distribution of reward discrepancy after fine-tuning with EAPO, DPO and S-DPO algorithms.

- HitRatio@5 (HR@5): Under the same candidate set configuration, this metric evaluates whether the true target item appears among the top 5 results recommended by the model. This metric assesses the model's recall performance under more relaxed conditions, better approximating user experience in practical recommendation application scenarios.

- NDCG@5: Normalized Discounted Cumulative Gain (NDCG) is an evaluation metric that considers the importance of ranking positions, assigning higher weights to correct recommendations that appear higher in the ranking. This metric comprehensively evaluates both the accuracy and ranking quality of recommendation results, with particular attention to the ranking effects of the top 5 positions, providing a more holistic reflection of the practical value of recommender systems.

Given that large language models inherently tend to generate singular, deterministic outputs rather than ranked lists, we adopted beam search techniques(Sutskever et al., 2014; Lei et al., 2023) to produce multiple recommendation candidates, thereby enabling evaluation under traditional ranking metrics. During the inference phase, we leveraged the beam search algorithm to obtain multiple potential recommendation results from the language model. Specifically, we set the beam width to k (where k=5 in this study), allowing the model to retain the k most probable output sequences at each decoding step. This approach enables the model to explore multiple potential recommendation paths rather than being constrained to a single output under a greedy decoding strategy. We ranked these k outputs according to the probability scores assigned by the model to each candidate sequence, thus forming an ordered recommendation list. The top 5 items from this list were then used to compute the Hit@5 and NDCG@5 metrics.

## C SUPPLEMENTARY EXPERIMENTS

### C.1 ANALYSIS OF REWARD DIFFERENCE DISTRIBUTION

Figure 3 illustrates the reward difference distributions for DPO, S-DPO, and EAPO algorithms across the training dataset. Here, reward difference is defined as the statistical distribution of model-predicted reward gaps between chosen items and each rejected item. Examining the distributional characteristics, we observe that standard DPO and S-DPO exhibit reward differences predominantly concentrated within a limited range (values primarily distributed between 1 and 3), indicating that these methods have constrained capability to quantify preference intensity and ineffectively differentiate subtle preference variations among different rejected items. In contrast, EAPO generates reward differences with a broader distributional range and larger mean values. This characteristic not only widens the decision boundary between chosen and rejected items, enhancing the model's discriminative power for positive samples, but also precisely quantifies the hierarchical preference differences

among various rejected items. Consequently, the model captures more fine-grained user preference information, providing richer discriminative signals for subsequent recommendation decisions.

## C.2 ANALYSIS OF COLD-START AND CROSS-DOMAIN SCENARIOS

To evaluate the robustness and generalization ability of EAPO, we conducted supplementary experiments under cold-start and cross-domain scenarios.

**Cold-Start Scenarios.** For the cold-start evaluation, we constructed two challenging scenarios using the Movies and TV dataset: a user cold-start scenario, where users in the test set are absent from the training set, and an item cold-start scenario, where items in the test set are unseen during training. The model was pre-trained on the training set and then evaluated on the corresponding test sets.

The results demonstrate EAPO's exceptional robustness in cold-start scenarios. Specifically, compared to the conventional warm-start setting, EAPO exhibits the minimal performance degradation in both user and item cold-start scenarios, with its HR@1 metric declining by only 4.93% and 7.20%, respectively. In contrast, the baseline S-DPO experiences drops of 6.57% and 2.07%. In the item cold-start setting, EAPO outperforms S-DPO by 12.74%, 5.96%, and 9.10% on HR@1, HR@5, and NDCG@5, respectively, indicating its superior ability to handle recommendations for unseen items. Furthermore, EAPO maintains a significant advantage over the traditional method, ChatRec, across all cold-start conditions, validating the effectiveness of our expert-guided preference optimization strategy under data sparsity.

Table 3: Performance comparison in conventional (warm-start) and cold-start scenarios.

| Scenario | Method | HR@1 | HR@5 | NDCG@5 |
|---|---|---|---|---|
| Conventional (Warm) | **EAPO** | **0.5760** | **0.6977** | **0.6331** |
| | S-DPO | 0.4841 | 0.6597 | 0.5781 |
| | ChatRec | 0.2271 | 0.4419 | 0.3056 |
| User Cold-start | **EAPO** | **0.5476** | **0.6823** | **0.6230** |
| | S-DPO | 0.4523 | 0.6423 | 0.5602 |
| | ChatRec | 0.2172 | 0.4420 | 0.3005 |
| Item Cold-start | **EAPO** | **0.5345** | **0.6700** | **0.6066** |
| | S-DPO | 0.4741 | 0.6323 | 0.5560 |
| | ChatRec | 0.2091 | 0.4384 | 0.2985 |

**Cross-Domain Scenario.** We further evaluated EAPO on the public Last.fm dataset from the music recommendation domain to assess its effectiveness in a new, non-e-commerce context. The cross-domain results further substantiate EAPO's generalization capabilities. Notably, EAPO's performance on the HR@1 metric is particularly outstanding, showing a 13.27% improvement over the runner-up method, S-DPO. This highlights EAPO's significant advantage in top-1 hit rate for music recommendation, a metric crucial for user experience. The ability of EAPO to remain competitive in a completely different application domain underscores the domain-agnostic nature and strong generalization power of the expert-assisted preference optimization framework.

Table 4: Performance comparison in the cross-domain scenario (Last.fm dataset).

| Method | HR@1 | HR@5 | NDCG@5 |
|---|---|---|---|
| **EAPO** | **0.7182** | **0.8116** | **0.7436** |
| S-DPO | 0.6341 | 0.8070 | 0.7091 |
| SASRec | 0.3486 | 0.7818 | 0.5768 |

## C.3 DISCRIMINATIVE CAPABILITY FOR HIGH-PREFERENCE NEGATIVE SAMPLES

Figure 4 presents robustness test results for preference-optimization-based recommendation algorithms when confronted with interference from high-preference negative samples. High-preference

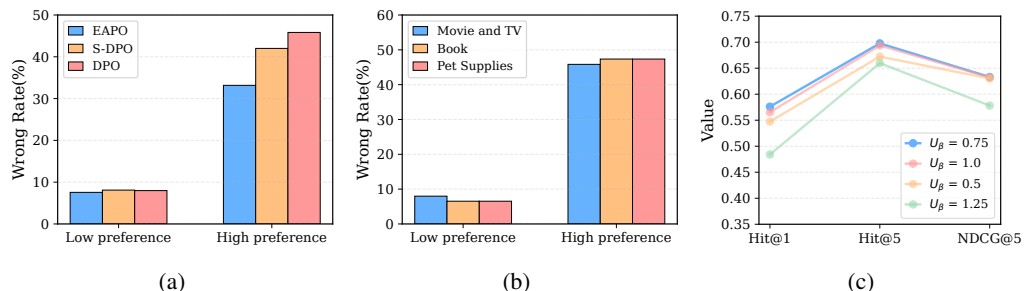

(a)                          (b)                          (c)

Figure 4: **(a)** & **(b)** Performance comparison between EAPO and baseline in the case of interference with high-preference negative samples. **(c)** Performance comparisons with varying values of $U_\beta$.

negative samples refer to candidate items that closely resemble users' true preferences but actually do not meet their needs, posing more challenging disturbances to recommendation systems. Experimental results demonstrate that, among model prediction errors, EAPO reduces the interference rate from high-preference negative samples by 12.3% compared to DPO and by 8.7% compared to S-DPO, highlighting its superior robustness and resistance to interference. These findings validate our theoretical hypothesis: by precisely adjusting the reward difference gradient between high-preference negative samples and positive samples, EAPO effectively enhances the model's ability to recognize subtle preference boundaries, thereby exhibiting higher discriminative accuracy and stability in complex recommendation scenarios.

### C.4    IMPACT OF $U_\beta$

In EAPO, $U_\beta$ is a critical hyperparameter that controls the normalized scaling range of $\beta_{(y_p, y_d)}$. Specifically, higher $U_\beta$ values indicate greater gradient magnitude differences between high-preference and low-preference samples during model parameter updates, while lower values indicate smaller differences. As shown in Figure 4c, recommendation performance initially increases and then decreases as $U_\beta$ increases, indicating that excessively large preference differences impede the model's effective learning of ranking relationships between samples, while excessively small preference differences fail to achieve effective sample differentiation. Based on experimental results, we set $U_\beta$ to 0.75 to achieve the optimal balance between sample ranking learning and sample differentiation capability.

### C.5    GENERALIZABILITY ANALYSIS

The proposed Adaptive Preference Parameter $\beta$ method demonstrates high generalizability by design, enabling direct integration into various existing pairwise preference optimization frameworks. To validate this extensibility, we applied the adaptive parameter mechanism to three representative preference optimization algorithms: IPO, CPO, and S-DPO, thereby developing their adaptive variants—EA-IPO, EA-CPO, EAPO and EA-SDPO. In our experimental design, we employed a consistent base model architecture and training corpus, introducing the adaptive parameter $\beta$ solely in the optimization objective function to ensure fair comparison. Figure 5 illustrates the reward function growth curves during training for each algorithm. The experimental results reveal that all adaptive variants exhibit significantly higher reward growth rates compared to their baseline counterparts, particularly during the early training stages. From a theoretical perspective, introducing the adaptive parameter $\beta$ fundamentally adjusts the balance between optimizing preference contrasts and maintaining language modeling capabilities. While traditional fixed-parameter methods struggle to adapt to variations across different training phases and data distributions, our adaptive mechanism dynamically adjusts the optimization direction based on the current training state, thus more effectively capturing subtle user preferences while preserving the model's generative capabilities. These experimental results robustly confirm that our proposed adaptive preference parameter method possesses strong algorithmic compatibility and consistent performance enhancements. In future work, we plan to further explore the integration of adaptive parameters with other optimization techniques, as well as investigate their potential applications in multimodal recommendation scenarios.

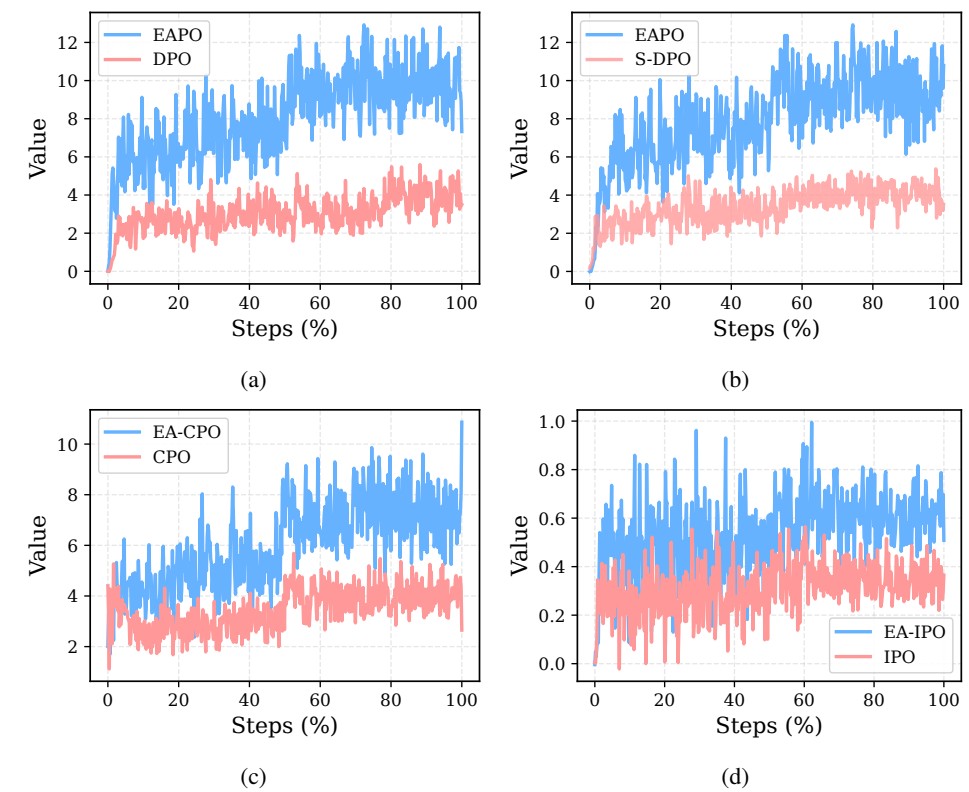

Figure 5: Comparison of EA-based reward margins among IPO, DPO, DPO and S-DPO algorithms..

# D    THE USE OF LLMS

We use LLMs to assist with language editing and polishing in the preparation of this manuscript. The primary purpose of the LLM was to improve the overall readability, clarity, and grammatical correctness of the text. We acknowledge that the LLM served solely as a language editing tool; all scientific contributions, including but not limited to the formulation of research ideas, method development, and interpretation of results, are entirely our own.

