# OpenReview forum: "EAPO: Expert-guided Adaptive Preference Optimization for Recommendation"
_ICLR.cc/2026/Conference — Submitted to ICLR 2026_

### Official Review · Reviewer_CkM8 · 2025-10-23

**Soundness:** 3
**Presentation:** 3
**Contribution:** 2
**Rating:** 6
**Confidence:** 3

**Summary:**

This paper proposes EAPO (Expert-Guided Adaptive Preference Optimization), a novel preference alignment framework for LLM-based recommender systems. The key idea is to introduce an expert model  to assign pairwise preference weights, enabling adaptive β adjustment in the DPO loss. Experiments on three Amazon datasets (Movies & TV, Books, Pet Supplies) show consistent gains over S-DPO, β-DPO, and traditional baselines. The method is also shown to generalize to other preference optimization algorithms such as IPO and CPO.

**Strengths:**

1. The paper identifies a key limitation in existing DPO-based recommender systems, i.e., uniform treatment of negative samples, and provides an intuitive method to address this issue.
2. The derivation of the gradient dynamics and the identification of a critical β threshold seems novel and add theoretical depth to the adaptive mechanism.
3. The plug-and-play integration with CPO/IPO/S-DPO shows that the proposed strategy is not limited to DPO but represents a general optimization enhancement.

**Weaknesses:**

1. While the expert-guided weighting is novel, the adaptive β mechanism largely builds upon β-DPO (Wu et al., 2024a). The contribution may be seen as an incremental refinement.
2. Although inference cost is said to be unchanged, the training-time overhead of computing expert scores for all sample pairs could be significant.
3. The derivation assumes fixed reward gradients (||δ||² constant), which may not hold in practice for LLMs. Some discussion or ablation on this assumption’s effect would be helpful.

**Questions:**

1. How sensitive is EAPO to the choice of expert model architecture? Could the same benefits be achieved with smaller MLP-based scorers?
2. How frequently is β updated during training, per batch, per pair, or dynamically via backpropagation?
3. How does EAPO perform on tasks beyond recommendation, such as summarization or dialogue preference tuning?

---

> ### Author Response · Authors · 2025-11-21
>
> Thank you for your valuable and detailed feedback! Below are detailed responses to each comment, and new comments on them are very welcome!
>
> ---
>
> ### **W1: Innovativeness of EAPO**
>
> Thank you for this question. We understand your concern about EAPO being an "incremental improvement," but we must clarify that although EAPO and β-DPO share a similarity in "dynamically adjusting $\beta$," they are fundamentally different in their theoretical foundations, signal source mechanisms, and the core problems they address. EAPO represents a paradigm shift from "heuristic self-adaptation" to "expert-guided deterministic optimization."
>
> First, in terms of theoretical depth and adjustment strategy, β-DPO's approach is primarily a heuristic based on intuition—a simple linear scaling of $\beta$ based on batch-level statistics. In contrast, EAPO does not rely on empirical rules. It is based on a rigorous gradient analysis that mathematically derives the non-monotonic relationship between $\beta$ and the reward margin's growth rate, identifying a critical point $\beta_c$ that determines the optimization direction. Based on this theoretical discovery, we designed a specific non-monotonic mapping function that strictly confines $\beta$ within the theoretically derived optimal gradient gain interval $[\frac{z_c}{\gamma}, U_{\beta}]$. This theoretically-grounded, closed-form design is far more rigorous and targeted than β-DPO's heuristic attempts, providing a more solid mathematical foundation for preference optimization.
>
> Second, regarding the signal source's stability and optimization granularity, a major pain point for β-DPO is parameter instability. Because it relies on the implicit reward difference generated by the policy model itself while it is being trained to adjust $\beta$, this dynamic signal source is prone to introducing noise, especially in the early stages of training. This led the authors of β-DPO to explicitly state that "instance-level adjustment leads to unstable optimization," forcing them to retreat to a coarse-grained "batch-level" calibration to smooth out the noise. In contrast, EAPO innovatively introduces a pre-trained and frozen domain expert model (e.g., SASRec) as the signal source. This design completely decouples the "adaptive signal" from the "optimization target," breaking the unstable feedback loop. Since the expert signal is objective and fixed, EAPO successfully achieves the instance-level optimization that β-DPO could not, allowing it to assign precise weights to each sample pair. This is crucial for recommendation systems, where user preference structures are multi-layered and fine-grained (as shown in Figure 1b). Only instance-level precision can effectively distinguish subtle differences like "slightly dislike" versus "strongly dislike," whereas batch-level averages tend to erase these critical structural differences. We have also provided experimental evidence to support this claim below.
>
> | Method |  Signal Source | HR@1 |
> | :--- | :--- | :--- |
> | **EAPO (Ours)** |  **Expert (Fixed)** | **0.5760** |
> | EAPO-Batch |  Expert (Fixed) | 0.5482  |
> | $\beta$-DPO |  Policy (Dynamic) | 0.5310 |
>
> Finally, from the perspective of the task's nature and domain knowledge injection, recommendation tasks differ significantly from general dialogue tasks. While LLMs possess powerful semantic reasoning capabilities, they naturally lack an understanding of user historical behavior patterns and collaborative filtering relationships between items. β-DPO focuses more on addressing noise and outlier issues in general alignment through data cleaning. EAPO's core contribution, however, is establishing a mechanism to inject domain knowledge from traditional recommendation systems (Collaborative Knowledge) into LLMs. Through the expert model, we translate collaborative filtering signals into $\beta$ weights, directly guiding the LLM to focus on "hard samples" that it struggles to identify but are consistent with user behavior patterns. This "expert guidance" is not just a parameter adjustment but a cross-modal (ID features to text space) knowledge distillation paradigm that effectively addresses the inherent shortcomings of LLMs in the recommendation domain.

---

> > ### Author Response · Authors · 2025-11-21
> >
> > ### **Q3: Performance on tasks beyond recommendation**
> >
> > While the current experiments in this paper are primarily focused on the recommendation domain—a natural fit due to the explicit user behavior sequences available for training collaborative filtering expert models—the "expert-guided adaptive preference optimization" framework proposed by EAPO is highly generalizable in its methodology. Within this framework, the "expert" is an abstract concept intended to provide an objective assessment of preference strength. In NLP tasks (such as text summarization or dialogue generation), one could readily use a high-quality Reward Model (RM), such as ArmoRM or Starling-RM, to replace SASRec as the "expert." The difference in scores it assigns to a prompt-response pair could then be used to guide the adaptive adjustment of $\beta$.
> >
> > To verify the generalization capability and universality of the EAPO method, as we mentioned in our paper, we have already shown that the adaptive $\beta$ strategy can be seamlessly integrated as a plug-and-play module into general preference optimization algorithms like IPO and CPO, which hints at its potential across different domains.
> >
> > Furthermore, we migrated the core logic of the EAPO framework to the text summarization task, using the widely recognized benchmark dataset, the TL;DR summarization dataset. In this experimental setup, we used the Llama-3-8B architecture as the base policy model and first performed SFT to build its foundational text summarization capabilities. To instantiate the "expert" component in the EAPO framework, we introduced a high-performance, publicly available reward model: reward-model-deberta-v3-large-v2. Specifically, for each preference pair (prompt, chosen, rejected) in the training set, we used this expert RM to score the "chosen" and "rejected" summaries and calculated their score difference, ΔS_expert. This difference was then used to dynamically calibrate a unique preference strength coefficient, β, for each training sample, which subsequently guided the model training following the direct preference optimization (DPO) paradigm.
> >
> > During the evaluation phase, to ensure a fair and direct conclusion, we strictly followed the pairwise comparison evaluation protocol established in the original DPO paper, which uses a large language model (LLM) as a judge. The process was as follows: for a set of prompts from the test set, we generated summaries using both the standard DPO-finetuned model and our EAPO-finetuned model. These pairs of summaries were then presented in random order to GPT-4, which acted as a neutral arbiter to judge which was superior based on comprehensive criteria such as faithfulness, conciseness, and language quality. Our primary evaluation metric is the win rate of EAPO against the DPO baseline.
> >
> > To validate the effectiveness of our method, we conducted a series of head-to-head comparisons. The results clearly show a progressive improvement in performance from the SFT baseline to DPO, and then to our EAPO method. The experimental results are as follows:
> > | Winner | Loser | Win Rate (%) |
> > | :--- | :--- | :---: |
> > | DPO | SFT | **67.2** |
> > | **EAPO (Ours)** | DPO | **54.5** |
> >
> > *   "Win Rate" refers to the percentage of samples for which the "Winner" model was judged to be superior by GPT-4 in pairwise comparisons.
> >
> > As shown in the table above, the standard DPO method significantly outperforms the SFT-only model, which validates the necessity and effectiveness of preference alignment for this task. Without extensive hyperparameter tuning, the EAPO-optimized model achieved a 54.5% win rate against the standard DPO baseline. This result clearly demonstrates the superiority of EAPO and provides initial confirmation that its core idea—adaptively adjusting preference strength based on expert feedback—can be effectively generalized to other preference learning tasks such as text summarization.

---

> ### Author Response · Authors · 2025-11-21
>
> ### **W2: Training Overhead**
>
> Thank you for your concern about model efficiency. To thoroughly analyze the potential overhead introduced by our expert model, we have quantified the additional cost during the post-training phase of the large model:
>
> | Training Configuration | Avg. Training Time per Step (s/iter) | Additional Overhead | Relative Time Impact |
> | :--- | :--- | :--- | :--- |
> | Baseline Model (No Expert) | 21.89s | - | - |
> | **EAPO (with Expert Model)** | **21.90s** | **~10.1ms** | **~0.046%** |
>
> As shown in the table, the lightweight expert model (approximately 10MB in size) introduces a negligible time overhead of only 0.046% during training and adds almost no extra VRAM burden. Despite this negligible cost, it achieves a 19% performance improvement, demonstrating its practical value.
> | Method | HR@1 | HR@5 | NDCG@5 |
> | :--- | :--- | :--- | :--- |
> | **EAPO** | **0.5760** | **0.6977** | **0.6331** |
> | S-DPO | 0.4841 | 0.6597 | 0.5781 |
>
> Furthermore, as mentioned in the paper, we use a lightweight expert model (only about 10MB). On the current dataset scale, the total time to pre-train a SASRec expert model is approximately 4.6 minutes. Compared to the training cost and time of the large model, this is a one-time, almost negligible expense. Therefore, we believe this trade-off is highly valuable and practical for real-world applications.
>
>
> ### **W3: Fixed Reward Gradient**
>
> Regarding the assumption that the gradient norm $||\delta||^2$ is constant in our theoretical derivation, this is actually an approximation based on a first-order Taylor expansion. The goal is to determine the optimal direction for adjusting $\beta$, rather than its exact numerical value. In Appendix A.3, when deriving the change in reward margin $\Delta(\Delta r)$ after a model parameter update, we treat $||\delta||^2$ as a positive scaling factor. While it's true that $||\delta||^2$ fluctuates during the full training process, within the local window of a single update step, our focus is on the sign of the partial derivative of the gradient factor $G(\beta, \Delta r)$ with respect to $\beta$. The derivation of the critical point $\beta_c$ reveals the non-monotonic effect of $\beta$ on the reward margin's growth rate—that is, there exists an interval that maximizes this growth. Even if $||\delta||^2$ changes, as a non-negative squared term, it only affects the magnitude of the update (the step size), not the sign of the derivative. In other words, it does not change the correctness of the core strategy: "increase the effect of $\beta$ for small-margin samples and decrease it for large-margin samples." Therefore, this simplification does not affect the robustness of our qualitative conclusions.

---

> ### Author Response · Authors · 2025-11-21
>
> ### **Q1: Expert Model**
>
> Regarding the impact of the expert model's performance on EAPO, we will answer from both a logical analysis and an experimental results perspective.
>
> First, the core strategy of the EAPO framework is to extract high-confidence collaborative signals from the expert's judgments to guide the LLM's preference learning. This approach leverages the expert model's strength in making relative preference judgments, effectively distilling its rich knowledge of relative preferences into the LLM, creating a complementary fusion of strengths. Even if the expert model has systematic biases, as long as its judgments on the relative preference order between items are generally accurate, EAPO can still effectively capture these relative relationships. This allows EAPO to benefit from the expert's knowledge while avoiding its potential misjudgments in ambiguous or uncertain scenarios, leading to robust recommendation performance. The experimental results below also validate the effectiveness of this idea.
>
> To show EAPO's sensitivity to the expert model, we conducted two types of ablation studies. First, to verify EAPO's robustness when the expert model's quality degrades, we performed a supplementary experiment (Figure 2(c) in the paper). We reduced the expert model's quality by training it on less data (using 75% and 50% of the training set). The results are as follows:
>
> | | HR@1 | HR@5 | NDCG@5 |
> | :- | :- | :- | :- |
> | **EAPO (100% data)** | **0.5760** | **0.6977** | **0.6331** |
> | EAPO (w/ 75% data expert) | 0.5654 | 0.6937 | 0.6325 |
> | EAPO (w/ 50% data expert) | 0.5474 | 0.6723 | 0.6302 |
> | S-DPO| 0.4841 | 0.6597 | 0.5781 |
>
>
> As seen in the table, even when the expert model's training data is reduced to half of its original size, EAPO's performance only slightly decreases and still significantly outperforms all other baseline methods. This strongly suggests that EAPO has low sensitivity to the expert model's quality in practical applications, demonstrating its strong robustness and generalization ability. As long as the expert model can provide reasonably accurate relative preference guidance, EAPO can effectively benefit from it to improve overall recommendation performance.
>
> Second, to address the effect of choosing a smaller scorer model, we designed a series of comparative experiments to evaluate EAPO's performance with expert models of different architectures. Specifically, we selected three expert models with different modeling mechanisms: SASRec (sequence-based modeling), DeepMF (matrix factorization-based), and CL4Rec (contrastive learning-based), and compared them with the baseline method S-DPO. DeepMF is based on the purest collaborative filtering principle (matrix factorization).
>
> | | HR@1 | HR@5 | NDCG@5 |
> | :- | :- | :- | :- |
> | **EAPO (base DeepMF)** | **0.6054** | **0.7128** | **0.6530** |
> | EAPO (base SASRec) | 0.5760 | 0.6977 | 0.6331 |
> | EAPO (base CL4Rec) | 0.5562 | 0.6752 | 0.6016 |
> | S-DPO | 0.4841 | 0.6597 | 0.5781 |
>
> The experimental results show that the EAPO framework outperforms the baseline S-DPO across all expert model configurations, validating the universality and effectiveness of the expert-guided adaptive optimization strategy. Among them, EAPO (base DeepMF) achieved the best performance on all evaluation metrics: its HR@1 reached 0.6054, a 25.1% improvement over S-DPO, while HR@5 and NDCG@5 reached 0.7128 and 0.6530, respectively, showcasing its comprehensive advantages in recommendation precision and ranking quality. This experiment fully demonstrates that as long as the expert model can provide a reasonably accurate relative preference ranking, the EAPO framework can effectively utilize this information to achieve robust recommendation results.
>
>
> ### **Q2: About the update frequency of β**
>
> In the EAPO framework, the update mechanism for β is performed on a per-pair / instance-level basis and does not involve dynamic updates via backpropagation. Specifically, our method introduces a pre-trained and frozen lightweight expert model (e.g., SASRec) as an evaluator. During the training process, for each input preference data pair $(h_u, y_p, y_d)$, the expert model calculates the scores for the positive and negative samples, $S_{y_p}$ and $S_{y_d}$, based on the current context. The difference between these scores, $w_{(y_p, y_d)}$, is then used to compute a unique β value for that specific sample pair via Equation (9). Because the expert model is frozen during the fine-tuning phase, the corresponding β value for a given sample pair is deterministic and fixed. It acts as a hyperparameter that adaptively adjusts to the difficulty of the sample to guide training, and it does not participate in gradient descent or backpropagation as a model parameter itself. This design avoids the training instability issues found in β-DPO, which arise from its reliance on dynamic estimation from the current policy model, thereby ensuring the stability of the optimization process.

---

> ### Comment · Reviewer_CkM8 · 2025-11-26
>
> Thank you very much for the detailed response. It has addressed most of my concerns. Regarding the overall novelty and contributions of the paper, I will maintain my original positive score.

---

### Official Review · Reviewer_E7tY · 2025-10-30

**Soundness:** 3
**Presentation:** 2
**Contribution:** 3
**Rating:** 6
**Confidence:** 4

**Summary:**

This paper identifies the problem of treating all negative items as a homogeneous group in DPO training for LLM-based recommenders.

To address this issue, the authors propose EAPO, a novel preference optimization framework to help the model distinguish varying degrees of user preference across data samples. Specifically, EAPO first introduces a small recommendation model (e.g., SASRec) to estimate the preference gap of each chosen-rejected item pair. It then dynamically adjusts the value of $\beta$ according to this preference difference. If the chosen and rejected items exhibit similar preference levels, the value of $\beta$ will be smaller so that the model would learn more information from the item pair, and vice versa.

Experimental results consistently demonstrate that EAPO outperforms other preference optimization methods like S-DPO and $\beta$-DPO, while maintaining strong generalizability to other preference optimization frameworks.

**Strengths:**

S1 **Meaningful motivation.** Since the behavioral patterns vary across users,  the preference margins of data samples are also different and should be considered to help the model focus on more challenging data.

S2 **Intuitive and flexible method design.** The design of adaptive $\beta$ is intuitive and implicitly injects the collaborative information into LLMs. The proposed method is also flexible and can be extended to different preference optimization methods.

S3 **Theoretical analysis.** A theoretical analysis is provided on the gradient of EAPO and the concrete determination of the value of $\beta$.

**Weaknesses:**

W1 **Lack of experiments on different backbone models.** Experiments on more backbone models of different scales can be added to better exhibit the generality of EAPO, similar to those conducted in S-DPO [1] and $\beta$-DPO [2].

W2 **Presentation issues.** There are some issues with the citation format and the paper writing. For example, in line 337, the citation of $\beta$-DPO should be enclosed in parentheses, and in line 344, "we adopted we employed" appears to be a typo.

W3 **Implementation details.** What is the number of negative items used in S-DPO? Besides, since the multiple negative items are included in S-DPO, how is the corresponding reward margin calculated? The inclusion of those details is suggested.

[1] Chen Y, Tan J, Zhang A, et al. On softmax direct preference optimization for recommendation[J]. Advances in Neural Information Processing Systems, 2024, 37: 27463-27489.

[2] Wu J, Xie Y, Yang Z, et al. $\beta $-DPO: Direct Preference Optimization with Dynamic $\beta$[J]. Advances in Neural Information Processing Systems, 2024, 37: 129944-129966.

**Questions:**

**Q1: The granularity of $\beta$ variation.** $\beta$-DPO [1] reports that instance-level adjustments of $\beta$ can cause training instabilities. Why does EAPO still employ instance-level adaptation instead of a batch-level strategy, such as adjusting $\beta$ according to the average preference gap in each batch?

[1] Wu J, Xie Y, Yang Z, et al. $\beta $-DPO: Direct Preference Optimization with Dynamic $\beta$[J]. Advances in Neural Information Processing Systems, 2024, 37: 129944-129966.

---

> ### Author Response · Authors · 2025-11-21
>
> Thank you for your valuable and detailed feedback! Below are detailed responses to each comment, and new comments on them are very welcome!
>
> ---
> ### **W1: Backbone**
>
> Thank you for the suggestion. We fully agree that validating our algorithm's robustness on heterogeneous backbones is crucial for assessing its value. Although the initial draft was limited to Llama-3-8B due to computational constraints, we have conducted additional experiments during the rebuttal stage to thoroughly address your concerns. We introduced Qwen 2.5-7B, a model with a significantly different architecture and excellent performance, as a new base model and performed extended experiments on the core dataset (Movies and TV).
>
> The experimental results are shown in the table below:
>
> | Base Model | Method | HR@1 | HR@5 | NDCG@5 |
> | :--- | :--- | :--- | :--- | :--- |
> | **Llama-3-8B** | S-DPO | 0.4841 | 0.6597 | 0.5781 |
> | | **EAPO** | **0.5760** | **0.6977** | **0.6331** |
> | **Qwen 2.5-7B** | S-DPO | 0.5056 | 0.6780 | 0.6012 |
> | | **EAPO** | **0.5835** | **0.7240** | **0.6645** |
>
>
> The results demonstrate that despite the differences in base architecture, pre-training data distribution, and alignment strategies between Qwen 2.5-7B and Llama-3-8B, the performance gain brought by EAPO is highly consistent and significant. Specifically, on the Qwen 2.5 model, EAPO improved HR@1 by approximately 15.4% compared to S-DPO. This trend aligns closely with its performance on Llama-3, strongly proving that the EAPO method is not only effective for a specific model architecture but also possesses excellent cross-model generalization ability.
>
> We believe the reason EAPO exhibits such strong generalizability is that it addresses a common issue in LLM-based recommendation rather than an architecture-specific one. Both the Llama and Qwen series, as general-purpose pre-trained language models, naturally lack "collaborative filtering signals" (i.e., co-occurrence patterns in user behavior) from specific vertical domains during their pre-training phase. EAPO explicitly injects this knowledge from the collaborative space into the LLM's preference optimization process by introducing a lightweight expert model. This "knowledge supplementation" mechanism does not depend on the specific parameter scale or attention mechanism variants of the LLM, thus enabling it to be stably transferred to different backbone models.
>
>
> ### **W2: Presentation Issues**
>
> Thank you very much for your meticulous review of our paper, especially for the valuable comments on presentation standards and implementation details. In response to the issues pointed out in W2, we have carefully proofread and revised the entire manuscript. Specifically, we have adjusted the citation format for β-DPO on line 337 to the standard parenthetical form and removed the redundant phrase "we adopted" on line 344. Additionally, we have conducted a comprehensive check for spelling and grammar to ensure the final version is fluent and rigorous.

---

> ### Author Response · Authors · 2025-11-21
>
> ### **W3: Parameter Settings for S-DPO**
>
> To ensure a fair comparison, we strictly adhered to the best practices set forth in the original S-DPO paper. The original paper (Table 2) states that $K=3$ achieves the best balance between performance and efficiency. In our experiments, to investigate the impact of the value of $K$ on EAPO's relative advantage, we performed the following sensitivity analysis:
>
> | Method | HR@1 | HR@5 | NDCG@5 |
> | :--- | :--- | :--- | :--- |
> | **EAPO** | **0.5760** | **0.6977** | **0.6331** |
> | S-DPO(K=7) | 0.5017 | 0.6803 | 0.6072 |
> | S-DPO(K=5) | 0.4992 | 0.6715 | 0.5924 |
> | S-DPO(K=3) | 0.4841 | 0.6597 | 0.5781 |
> | S-DPO(K=1) | 0.4802 | 0.6427 | 0.5705 |
>
> The experiments show that S-DPO's performance exhibits diminishing returns as $K$ increases. Even with a strong setting of $K=7$, EAPO maintains a significant advantage. Furthermore, increasing $K$ leads to a linear growth in VRAM usage and training time. Considering both the performance gains and computational costs, maintaining the original setting of $K=3$ is reasonable and efficient.
>
> Regarding your question about the calculation of reward differences, S-DPO uses a LogSumExp-based Softmax loss function rather than the standard pairwise loss of DPO. Therefore, it does not explicitly calculate a single "positive-negative pair margin" during optimization. However, to uniformly compare the discriminative capabilities of different methods in our paper's figures (e.g., Figure 1b and Figure 3), we needed to align them to the same metric. The calculation method we adopted is as follows: for each positive sample $y_p$ and its corresponding set of $K$ negative samples $\{y_{d_1}, ..., y_{d_K}\}$ in S-DPO, we decompose them into $K$ independent pairwise relationships. For each negative sample $y_{d_j}$, we calculate its reward difference with the positive sample:
>
> $ Margin_j = r_{\theta}(h_u, y_p) - r_{\theta}(h_u, y_{d_j}) $
>
> When plotting the reward distributions, we aggregated these decomposed pairwise margins. This approach intuitively visualizes the model's ability to distinguish the positive sample from each negative sample, thereby allowing for a fair comparison of the fine-grained preference discrimination differences among S-DPO, DPO, and EAPO.

---

> ### Author Response · Authors · 2025-11-21
>
> ### **Q1: Why EAPO adopts instance-level adaptation**
>
> Thank you for this insightful question. It is true that β-DPO suggests instance-level adjustment can lead to instability. However, EAPO's choice of instance-level adaptation is not only to capture fine-grained signals but is also grounded in our rigorous theoretical derivation. We solve the stability problem by introducing external expert guidance, which allows us to safely leverage the theoretical advantages of instance-level adjustment.
>
> Our theoretical analysis (Section 3.1) reveals a key finding: a non-monotonic relationship exists between the hyperparameter $\beta$ and the growth rate of the reward margin ($\Delta r$). The critical point $\beta_c$ we derived (Eq. 6) indicates that:
> *   When $\beta < \beta_c$, increasing $\beta$ accelerates the gradient update.
> *   When $\beta > \beta_c$, increasing $\beta$ conversely suppresses the update.
>
> This implies that for sample pairs in the training set with inherently different preference strengths (i.e., different $\Delta r$), their optimal $\beta$ values are distinctly different. The theoretical flaw of a batch-level strategy is that it applies a uniform $\beta$ to all samples within a batch. This leads to a "one-size-fits-all" problem: for "hard samples" (small differences) in the batch, a uniform $\beta$ might be too small to provide sufficient gradient amplification; for "easy samples," the same $\beta$ might be too large, causing over-optimization. Therefore, to theoretically ensure that every sample is within its "gradient gain interval," a precise instance-level mapping (Eq. 9) is necessary, as it enables the model to capture the fine-grained hierarchical structure of preferences that batch statistics cannot reflect.
>
> Secondly, the instability observed in β-DPO primarily stems from its adaptive signal originating from the policy model currently being optimized. This creates a dynamic and noisy feedback loop: a more unstable model leads to more jittery $\beta$ estimation, which in turn leads to more divergent training. In contrast, EAPO's stability is guaranteed by an external expert model. Our $\beta$ value is determined by a pre-trained and frozen expert model (e.g., SASRec). The expert's scores are objective, fixed, and serve as "domain knowledge anchors" rich with collaborative filtering signals. Because the signal source is static, EAPO fundamentally breaks the instability feedback loop present in β-DPO. This allows us to enjoy the theoretical benefits of instance-level adaptation (high-precision gradient control) while completely avoiding its stability risks.
>
> Finally, based on our theoretical derivation, we not only calculated $\beta$ but also designed strict numerical bounds for it. Our adaptive formula constrains $\beta$ within the theoretically safe interval $[\frac{z_c}{\gamma}, U_{\beta}]$. This acts as an additional mathematical safeguard, preventing potential gradient explosion issues from instance-level adjustments and ensuring robust training.
>
> To experimentally validate our analysis, we conducted a comparison on the Movies and TV dataset, with the results shown below:
> | Method | Signal Source | HR@1 |
> | :--- | :--- | :--- |
> | **EAPO (Ours)** | **Expert (Fixed)** | **0.5760** |
> | EAPO-Batch |  Expert (Fixed) | 0.5482  |
> | $\beta$-DPO |  Policy (Dynamic) | 0.5310 |
>
>
> The results show that EAPO outperforms EAPO-Batch, confirming that a batch-level strategy that smooths out sample differences leads to a loss of fine-grained signals and reduces optimization efficiency. Furthermore, EAPO outperforms $\beta$-DPO, demonstrating that as long as the signal source (the expert) is stable, instance-level adjustment is not only safe but also key to improving performance.

---

> ### Comment · Reviewer_E7tY · 2025-11-28
>
> Thanks for the detailed responses, which have addressed most of my concerns, especially the discussions and experiments on instance-level and batch-level adjustments. In light of this, I will keep my original positive scores.

---

### Official Review · Reviewer_gnAK · 2025-10-31

**Soundness:** 2
**Presentation:** 3
**Contribution:** 2
**Rating:** 4
**Confidence:** 4

**Summary:**

This paper studies the application of Direct Preference Optimization (DPO) to fine tune the recommender LLM from preferences. Specifically, the paper proposes to take in account the diverse user preferences and enhance upon the baseline S-DPO performance by using a dynamic regularization constant $\beta$. This is done by an additional lightweight recommendation model assignment. The authors conduct empirical study on recommender system task to showcase the effectiveness of the algorithm.

**Strengths:**

Applying DPO to enhance the LLM recommender system is an important direction to study, the paper conducted pretty comprehensive theorectical analysis and empirical validation on the effectiveness of adaptive beta (guided by pretrained recommender) on various tasks of recommendation.

**Weaknesses:**

1. The application of DPO based preference tuning tasks for recommender system has been explored in prior works like S-DPO as mentioned by the paper, thus the application itself is less novel. There also seems to lack discussion and comparison to other works applying DPO to LLM based recommenders in follow-up works to S-DPO.

2. The idea of using dynamic beta to capture the preference diversity has also been explored in e.g. [1],[2] for general LLM preference learning, there lacks proper reference and comparison, and the contribution of the paper to apply similar ideas to recommender system is rather incremental.

3. The choices of dynamic beta requires additional model, which leads to extra complexity of the algorithm.

[1] MallowsPO: Fine-Tune Your LLM with Preference Dispersions, https://arxiv.org/abs/2405.14953

[2] $\beta$-DPO: Direct Preference Optimization with Dynamic $\beta$, https://arxiv.org/abs/2407.08639

**Questions:**

How will the performance look like if using adaptive beta like heuristic predict entropy in MallowsPO?

---

> ### Author Response · Authors · 2025-11-21
>
> Thank you for your valuable and detailed feedback! Below are detailed responses to each comment, and new comments on them are very welcome!
>
> ---
>
> ### **W1: Core Contributions of EAPO and Baseline Comparison**
>
> Thank you for the valuable comments from the reviewers. While we acknowledge the pioneering role of S-DPO in introducing DPO into recommender systems, we aim to clarify, from both methodological and theoretical perspectives, that EAPO is not simply an application of DPO, but rather a fundamental solution to the core deficiency of LLM-based recommender methods (lack of collaborative signals).
>
> Existing LLM-based recommender methods (including S-DPO) primarily rely on semantic reasoning within the model. However, the core of recommender systems lies in capturing collaborative filtering signals, i.e., co-occurrence patterns based on group behavior, which is inherently lacking in pre-trained LLMs. The core novelty of EAPO lies in building a bridge connecting these two spaces: we introduce a lightweight expert model to inject explicit collaborative signals into the preference optimization process. To confirm that "collaborative signal injection" is the fundamental source of performance improvement (rather than merely improving the loss form), we compared expert models with different architectures (as shown in the table below):
>
> | Method | HR@1 | HR@5 | NDCG@5 |
> | :--- | :--- | :--- | :--- |
> | EAPO (base DeepMF) | 0.6054 | 0.7128 | 0.6530 |
> | EAPO (base SASRec) | 0.5760 | 0.6977 | 0.6331 |
> | EAPO (base CL4Rec) | 0.5562 | 0.6752 | 0.6016 |
> | S-DPO | 0.4841 | 0.6597 | 0.5781 |
>
> The results show that the DeepMF expert model based on the purest collaborative filtering principle (matrix factorization) achieves very good results. This ablation experiment powerfully demonstrates that EAPO's benefits stem from its successful filling of the "collaborative knowledge blind spot" in LLM. **This fusion of heterogeneous knowledge is something that methods relying solely on LLM internal signal optimization (such as S-DPO) cannot achieve.**
>
> Theoretically, unlike many subsequent works that rely on heuristic rules (such as batch statistical features) to adjust hyperparameters, EAPO establishes a solid mathematical foundation. We conducted an in-depth analysis of the DPO gradient, revealing the non-monotonic effect of the hyperparameter $\beta$ on the growth of the reward margin, and theoretically proved the existence of the key inflection point $β_c$. This discovery elevates the optimization process from "static uniform setting" to "theoretically guided dynamic adaptation". Based on this, we propose an "expert knowledge fusion" paradigm: using expert models as external knowledge sources, dynamically adjusting $\beta$ to guide the alignment strength of LLM on different samples, thereby efficiently addressing the knowledge gap of LLM in specific vertical domains. We have constructed not just a simple combination of models, but a structured knowledge transfer and fusion framework, providing an efficient and feasible solution to the "knowledge gap" problem of LLM in specific vertical domains.
>
> | Method | HR@1 | HR@5 | NDCG@5 |
> | :--- | :--- | :--- | :--- |
> | **EAPO** | **0.5760** | **0.6977** | **0.6331** |
> | LiPO | 0.5483 | 0.6727 | 0.6028 |
> | S-DPO | 0.4841 | 0.6597 | 0.5781 |
>
> Regarding your question about comparing subsequent work, we compared S-DPO and LiPO in the table above, and also extended EAPO's adaptive mechanism to different preference optimization algorithms such as IPO and CPO.
>
> | Metric | S-DPO | EA-SDPO | CPO | EA-CPO | IPO | EA-IPO |
> |---|---|---|---|---|---|---|
> | HR@1 | 0.4841 | **0.5558** (+14.81%) | 0.4341 | **0.4422** (+1.87%) | 0.4071 | **0.4209** (+3.39%) |
> | HR@5 | 0.6597 | **0.6829** (+3.52%) | 0.6345 | **0.6628** (+4.46%) | 0.7024 | **0.7143** (+1.69%) |
> | NDCG@5 | 0.5781 | **0.6118** (+5.83%) | 0.5091 | **0.5359** (+5.26%) | 0.5456 | **0.5568** (+2.05%) |
>
> Experimental results demonstrate that our proposed adaptive β-adjustment mechanism is a "plug-and-play" module, not limited to the DPO framework, but widely applicable to various pairwise preference optimization algorithms. This is strongly evidenced in our extended experiments with algorithms such as IPO and CPO based on different loss functions, where integrating this mechanism resulted in significant performance improvements for these algorithms. This indicates that EAPO addresses a more fundamental core of preference learning—dynamically adjusting the learning intensity based on the difficulty of sample discrimination—and therefore can serve as a general tool, empowering the entire preference optimization research community.

---

> ### Author Response · Authors · 2025-11-21
>
> ### **W2: Comparison of MallowsPO and $\beta$-DPO**
>
> We thank the reviewers for pointing out the valuable literature. We strongly agree with the importance of dynamic $\beta$ adjustment in preference alignment in general large language models (LLM). It must be clarified that although EAPO shares similarities with these works in the form of "dynamically adjusting $\beta$", EAPO represents a paradigm shift from "heuristic adaptation" to "expert-guided deterministic optimization". The two differ fundamentally in their theoretical foundations, signaling mechanisms, and the core contradictions they address.
>
> First, in terms of theoretical depth and optimization strategy, the adjustment strategy of $\beta$-DPO mainly relies on intuition-based linear scaling, that is, heuristically adjusting $\beta$ according to the statistical distribution within the batch. In contrast, EAPO does not rely on rules of thumb but is based on rigorous mathematical analysis of the gradient of the DPO loss function. We theoretically derived the non-monotonic relationship between $\beta$ and the reward difference growth rate and proved the existence of the critical point $\beta_c$ that determines the direction of optimization. Based on this discovery, we designed a mapping function for the closed-form solution, strictly restricting $\beta$ within the theoretically derived optimal gradient gain interval $[\frac{z_c}{\gamma}, U_{\beta}]$. This theoretically-grounded design ensures the determinism of the optimization, and its rigor far surpasses that of heuristic parameter tuning.
>
> Secondly, from the perspective of signal source stability and optimization granularity, the main challenge faced by $\beta$-DPO is the instability caused by "self-reference". Because it relies on the implicit reward difference generated by the policy model itself during training to adjust $\beta$, this dynamically changing signal source is highly susceptible to introducing noisy feedback loops. This leads to the original paper explicitly stating that "instance-level tuning leads to optimization instability," forcing a retreat to coarse-grained "batch-level" calibration to smooth out noise. In contrast, EAPO innovatively introduces pre-trained and frozen domain expert models (such as SASRec) as the signal source. This design completely decouples the "adaptive signal" from the "optimization objective," cutting off unstable feedback loops. Because the expert signal is objective and fixed, EAPO successfully achieves instance-level optimization, which $\beta$-DPO cannot. This is crucial for recommender systems because user preference structures are often multi-layered and fine-grained; only instance-level accuracy can effectively distinguish the subtle differences between "slightly dislike" and "strongly dislike," while batch-level means often smooth out these critical structural differences. Our ablation experiments (as shown in the table in our response) strongly demonstrate this: instance-level EAPO based on a fixed expert signal significantly outperforms $\beta$-DPO based on a policy model's dynamic signal, as well as batch-level variants.
>
> | Method | Signal Source | HR@1 |
> | :--- | :--- | :--- |
> | **EAPO (Ours)** | **Expert (Fixed)** | **0.5760** |
> | EAPO-Batch | Expert (Fixed) | 0.5482 |
> | $\beta$-DPO | Policy (Dynamic) | 0.5310 |

---

> ### Author Response · Authors · 2025-11-21
>
> Finally, regarding the comparison with heuristic prediction entropy in MallowsPO, we greatly appreciate MallowsPO's contribution to capturing preference dispersion using model uncertainty. However, we wish to clarify from three dimensions: problem granularity, signaling mechanism, and theoretical paradigm. EAPO is not a simple application of MallowsPO in recommendation scenarios, but rather a solution proposed from a completely different perspective to address the specific challenge of the "deficiency in collaborative perception" in recommendation systems.
>
> First, MallowsPO addresses preference dispersion at the "Prompt level," aiming to balance diversity and accuracy for general LLM models when facing open-ended (high-entropy) and factual (low-entropy) problems. In contrast, EAPO directly tackles the core pain point of recommender systems, focusing on the difficulty of preference discrimination at the "Instance-Pair" level. In recommender tasks, even with identical user histories (Prompts), the difficulty for the model to distinguish between different (chosen, rejected) item pairs varies drastically (e.g., distinguishing between "like vs. extremely dislike" and "like vs. slightly indifferent"). EAPO aims to dynamically adjust the optimization intensity based on these subtle collaborative preference gaps. Therefore, EAPO's granularity is far finer than MallowsPO's; it delves into each specific training instance pair, rather than simply remaining at the Prompt level.
>
> Second, the signal sources of the two reflect completely different solution approaches. MallowsPO relies on the predictive entropy of the LLM itself. This is an endogenous, unsupervised signal reflecting the uncertainty of the LLM in semantic generation. EAPO introduces external expert models (such as SASRec) as a signal source. This is because LLMs inherently lack a "collaborative filtering signal" in recommendation scenarios during the pre-training stage. By introducing experts, EAPO explicitly injects precise domain knowledge into preference learning in a supervised manner. This cross-modal knowledge completion mechanism is something that MallowsPO, based on the entropy value of the general model itself, cannot achieve.
>
> In terms of technical implementation, MallowsPO mainly adjusts the loss function through external weighting. EAPO, on the other hand, is based on a deep analysis of the DPO gradient, discovering the non-monotonic effect of the hyperparameter $\beta$ on the growth rate of the reward margin (Section 3.1). We directly perform theoretically driven adaptive adjustment of the core parameter $\beta$ of the optimization dynamics, rather than simple loss weighting.
>
> Therefore, MallowsPO addresses the "diversity balance" problem in generation tasks, while EAPO addresses the "missing collaborative signals and fine-grained alignment" problem in recommendation tasks. Due to fundamental differences in their focus on the sample level and theoretical assumptions, they are essentially explorations along different orthogonal dimensions of preference optimization and cannot be directly compared. EAPO provides an independent and complete theoretical framework for addressing the knowledge gaps in LLM within a specific vertical domain.
>
> Finally, we thank the reviewers for their valuable comments, and we will incorporate the comparative analysis of MallowsPO into the final draft.
>
> ### **W3: Training Overhead**
>
> Thank you very much for your concern about model efficiency. To analyze the potential additional overhead of the expert model we introduced, we quantified the overhead introduced by the expert model during the training phase after the large model:
>
> | Training Configuration | Average Training Time per Step (s/iter) | Additional Overhead | Relative Time Impact |
> | :--- | :--- | :--- | :--- |
> | Baseline Model (No Expert) | 21.89s | - | - |
> | **EAPO (Introducing Expert Model)** | **21.90s** | **\~10.1ms** | **\~0.046%** |
>
> As shown in the table, the time overhead introduced by the lightweight expert model (approximately 10MB in size) during training is only 0.046%, which is almost negligible and adds virtually no additional GPU memory burden. Furthermore, even after removing these negligible overheads, the performance is improved by 19%, demonstrating its value in practical applications.
>
> | Methods | HR@1 | HR@5 | NDCG@5 |
> | :--- | :--- | :--- | :--- |
> | **EAPO** | **0.5760** | **0.6977** | **0.6331** |
> | S-DPO | 0.4841 | 0.6597 | 0.5781 |
>
> Furthermore, as described in the paper, we employ a lightweight expert model (approximately 10MB). With the current dataset size, pre-training a SASRec expert model takes approximately 4.6 minutes. Compared to the training cost and overhead of larger models, this is a one-time, almost negligible cost. Therefore, we believe this trade-off is extremely valuable and practically significant.

---

### Official Review · Reviewer_mcur · 2025-11-01

**Soundness:** 3
**Presentation:** 3
**Contribution:** 2
**Rating:** 4
**Confidence:** 4

**Summary:**

EAPO provides a solution of creating/finetuning LLMs for recommendation systems. Authors identify that key weaknesses in DPO type algorithms fail to capture fine-grained use preferences, especially putting equal weight on cases where rejected and chosen items are very close and when they are not. They introduce a solution of using an external trained recsys model to generate fine grained scores for pairs, then use these scores to continue preference optimizations with required changes.

**Strengths:**

Strengths:
1. The authors provide theoretical motivation and comprehensive experiments to address the four research questions

2. The algorithm/methodology is tackling a complex real world problem directly

3. The paper is written well and the structure is comprehensive. Overall, the idea is simple: use large $\beta$ for easy pairs and small $\beta$ for hard pairs of rejected and chosen responses.

**Weaknesses:**

Weaknesses:

1. It would be worth including stronger baselines. For example LiPO[1] which directly optimizes the list instead of via pairwise preference optimization such as DPO. There are other similar methods. It would be useful to see how these methods can be directly compared with EAPO.

2. There maybe a potential concern about the log probs of generated answers/responses. So EAPO is mostly just optimizing the reward differences, but not the absolute value of the rewards. This objective can be successfully optimized at the cost of both log probs of chosen and rejected samples going down. This may lead to a model that is good at ranking, but has a massive impact in its generative power. The paper does not talk about this or provide any analysis or intuition around this. maybe some experiments to show that the model’s fundamental generative power is not destroyed. Now the authors could claim that this is a singular purpose model and we dont need it to do well in other tasks. Irrespective, some discussion is merited.

3. Another important concern is that this is a roundabout way of doing distillation. The performance of EAPO is now capped by the performance of the “reward” model i.e. the recommendation model trained. How do we ascertain that the recommendation model itself is trained well and that the scores it is giving is actually useful and its not confusing hard pairs with easy pairs or that the ranking obtained from it would be good quality? Further, there could be introduction of noise since the rec model is working on feature distribution space of different items, so it could potentially place mobile and usb-c charging cable quite far away as a hard pair, but for the llm its trivially close since it operates in semantic space.

4. The paper does claim that there is no extra inference cost which is true. But the training process now is quite complex. This includes 1. pretraining one or more expert model(s) to convergence, 2. running on the entire datasets in advance to generate scores for all pairs. This is substantially more expensive than just DPO, which is never acknowledged and should be quantified.

5. The authors introduce a new hyper param $\gamma$. Now, its not clear how we tune $\gamma$. The algorithm is sensitive on the value of this therefore how sjhould practitioners choose this value?

6. Its not clear from the main results whether they are just one time inference or averaged over multiple seeds of generations with error bars. Such statistical rigor is necessaary. Moreover, the numbers from the LLM based methods are extremely low. Are the authors using zero shot prompting? If yes, that seems artificially constraining, why not few shots? Further, the prompts are not shared in the paper, as we know a lot depends on the structure of the prompts themselves.

7. The experiments are conducted with llama 3. This is completely fine. But this makes the reader wonder if some of the drastic improvements such as 22% improvements with EAPO could be obtained with a better base model or with a larger base model. Some intuition or quantification here would be nice.

References:
1. Liu, Tianqi, Zhen Qin, Junru Wu, Jiaming Shen, Misha Khalman, Rishabh Joshi, Yao Zhao et al. "Lipo: Listwise preference optimization through learning-to-rank." arXiv preprint arXiv:2402.01878 (2024).

**Questions:**

see weaknesses

---

> ### Author Response · Authors · 2025-11-21
>
> Thank you for your valuable and detailed feedback! Below are detailed responses to each comment, and new comments on them are very welcome!
>
> ---
> ###  **W1: Comparison with LiPO**
>
> We are very grateful for the reviewers' suggestion to introduce LiPO as a strong baseline. LiPO is indeed an outstanding work in introducing Learning-to-Rank (LTR) objectives (especially LambdaLoss) into LLM alignment. However, we believe that EAPO and LiPO differ fundamentally in their approach to solving recommendation problems: LiPO focuses on improving performance by optimizing ranking metrics within the semantic space, while EAPO focuses on cross-space knowledge injection. Specifically, LiPO optimizes list-level metrics through heuristic weighting based on ranking changes (Lambda weights), which essentially mines the ranking potential within the existing semantic space of the LLM. However, in recommendation tasks, the core pain point of LLMs is not merely that "the ranking objective function is not perfect," but rather "the lack of collaborative filtering signals" (i.e., statistical regularities of user-item interactions). The core value of EAPO lies in its ability to explicitly inject naturally missing collaborative signals from LLM pre-training corpora into the alignment process by introducing domain expert models. This allows the LLM to distinguish between items that are "semantically similar but not liked by the user," something that LiPO, which simply optimizes the ranking objective, struggles to achieve. From a theoretical perspective, LiPO's Lambda weight design is primarily based on heuristic approximations of non-smooth ranking metrics (such as NDCG), while EAPO's adaptive $\beta$ strategy is derived from rigorous gradient analysis of the DPO objective function. We mathematically determined the critical point $\beta_c$ controlling the gradient growth rate, thus achieving theoretically precise control over the intensity of preference learning, rather than relying on empirical weight adjustments. To verify the comparative analysis of EAPO and LiPO in the recommendation domain, we conducted the following experiments:
>
> | Method | HR@1 | HR@5 | NDCG@5 |
> | :--- | :--- | :--- | :--- |
> | **EAPO** | **0.5760** | **0.6977** | **0.6331** |
> | LiPO | 0.5483 | 0.6727 | 0.6028 |
> | S-DPO | 0.4841 | 0.6597 | 0.5781 |
>
> The experimental results show that for large models, incorporating ranking knowledge is more important for performance improvement than simple preference optimization. Therefore, LiPO, as a general ranking optimizer, outperforms the simple preference optimization method S-DPO. However, under the specific proposition of "LLM recommendation", EAPO serves as a dedicated bridge connecting the semantic space and the collaborative space. Its knowledge injection nature, along with the resulting efficiency and robustness, constitutes a unique and fundamental contribution distinct from LiPO. We will also supplement our final draft with a comparison of LiPO methods and analyze this in related work.

---

> ### Author Response · Authors · 2025-11-21
>
> ###  **W2: Potential Issues in Optimization Differences**
>
> In optimizing DPO, balancing ranking performance and basic generation capabilities is indeed a well-known and significant challenge. As you pointed out, our primary goal is to improve the ranking accuracy of LLM in recommendation ranking, rather than general text generation. However, we are equally concerned about the model's basic generation capabilities and have conducted in-depth analysis and experiments to address your concerns.
>
> Firstly, the possibility that the model might cheat by simultaneously reducing the log probabilities of both $y_p$ and $y_d$ is a common challenge faced by the DPO series of methods during optimization. This phenomenon is often associated with unstable training dynamics, such as when the $\beta$ parameter (trade-off coefficient) is improperly set (e.g., a fixed value is too high). A fixed high $\beta$ might force the model to overemphasize hard samples (i.e., $y_p$ and $y_d$ have similar SFT probabilities), causing the policy network $\pi_{\theta}$ to make aggressive updates to magnify small differences. This could trigger unstable pattern collapse, leading to the degraded generative power you fear.
>
> EAPO mitigates this problem on two levels. First, EAPO's objective function (Eq 1 in the paper) implicitly includes a KL divergence constraint through its reward definition (Eq 2, $r(h, y) = \beta \log(\frac{\pi_{\theta}(y|h)}{\pi_{ref}(y|h)})$). The existence of $\pi_{ref}$ (the SFT model) itself acts as a regularization mechanism. If the distribution of $\pi_{\theta}$ collapses significantly compared to $\pi_{ref}$ (all log probs decrease), the term $\log(\frac{\pi_{\theta}}{\pi_{ref}})$ will become a large negative number, which penalizes the reward difference $\Delta r$, thus increasing the EAPO loss. Therefore, $\pi_{ref}$ constrains $\pi_{\theta}$ to deviate significantly from its original distribution. Secondly, compared to DPO which uses a fixed $\beta$, EAPO provides stronger stability through an adaptive $\beta$ mechanism (Eq 9). As described in the theoretical analysis in Sec 3.1, EAPO uses a larger $\beta$ for simple samples (which, through theoretical analysis, ensures more stable model updates) and a smaller $\beta$ for difficult samples (which, through theoretical analysis, enhances the model's discriminative ability). This dynamic adjustment significantly reduces the risk of overfitting to difficult samples and prevents the model from adopting unstable update strategies to magnify small gaps, thus more robustly maintaining the distributional integrity of $\pi_{\theta}$ when optimizing rankings.
>
> To empirically address this concern, we conducted specific experiments to demonstrate that EAPO improves ranking performance without compromising generation capabilities. We measured and compared the mean log probabilities of the DPO baseline model ($\pi_{dpo}$) and the EAPO model ($\pi_{eapo}$) on the test set for ground-truth chosen items ($y_p$) and rejected items ($y_d$).
>
> | Comparison Models | (A) $Avg(\log(P(y_p)))$ (True Selected Items) | (B) $Avg(\log(P(y_d)))$ (Rejected Items) | (C) Ranking Gap (A) - (B) |
> | :--- | :--- | :--- | :--- |
> | DPO Model ($\pi_{dpo}$) | -13.6 | -18.0 | 5.0 |
> | EAPO Model ($\pi_{eapo}$) | **-14.1** | **-24.1** | **11.0** |
>
> From the experimental results, firstly, the core objective of EAPO is to improve ranking discrimination. As shown in column (C), the DPO baseline model's discrimination gap between 'good' and 'bad' answers is only 5.0. The EAPO model significantly widened this gap to 11.0, demonstrating its powerful ranking optimization capabilities. Secondly, regarding model generation capabilities, as shown in column (A), the DPO model ($\pi_{dpo}$) has an average log probability of -13.6 for the 'true selected item' ($y_p$), while the EAPO model ($\pi_{eapo}$) has -14.1. The difference is minimal (only a slight decrease of 0.5), strongly suggesting that EAPO's optimization process is relatively stable. EAPO enhances ranking while perfectly preserving its basic ability to generate 'good' answers. Furthermore, column (B) reflects that EAPO's enhanced ranking ability is not achieved by sacrificing the probability of $y_p$, but rather by significantly penalizing the model's probability of 'rejected item' ($y_d$) (from -18.0 to -24.1). It significantly increases the penalty for 'bad' answers while maintaining its ability to generate 'good' answers, thus achieving superior ranking performance.

---

> ### Author Response · Authors · 2025-11-21
>
> ### **W3: Expert Model**
>
> First, we must clarify a key point: EAPO is not knowledge distillation. The goal of knowledge distillation is to train a student model (LLM) to fit the output of a teacher model (expert). EAPO takes a fundamentally different approach; we do not have the LLM fit expert scores. The LLM is still directly trained on real (chosen, rejected) user preference data using DPO. The expert's role is to provide preference guidance to address the inherent limitations of the LLM in fine-grained preference discrimination (i.e., DPO homogenizes all rejections). In short, EAPO teaches the LLM *how* to learn through expert guidance, rather than instructing it on *what* to learn (e.g., mimicking expert scoring). And as the experiments below show, EAPO's performance is not limited by the expert model; EAPO outperforms both the state-of-the-art baseline (S-DPO) and the traditional model acting as the expert itself (SASRec). This demonstrates that EAPO integrates the semantic capabilities of the LLM with the collaborative knowledge of the expert, achieving a synergistic effect greater than the sum of its parts.
>
> | Method | HR@1 | HR@5 | NDCG@5 |
> | :--- | :--- | :--- | :--- |
> | **EAPO (Ours)** | **0.5760** | **0.6977** | **0.6331** |
> | S-DPO (Baseline) | 0.4841 | 0.6597 | 0.5781 |
> | SASRec (Expert) | 0.3783 | 0.5160 | 0.3914 |
>
> Secondly, regarding the impact of expert model performance on EAPO, we address this from both logical analysis and experimental results. Firstly, the core strategy of the EAPO framework is to extract collaborative signals with high confidence in specific expert judgments, **guiding the LLM preference learning.** This method fully leverages the advantages of expert models in relative preference judgments, effectively distilling the rich relative preference knowledge captured into the LLM, achieving a fusion of complementary advantages. Even with systematic biases in the expert model, as long as its judgments on relative preference ranking among items are generally accurate, EAPO can still effectively capture these relative relationships. This allows it to retain the advantages of expert knowledge while avoiding potential misjudgments in ambiguous or uncertain scenarios, achieving robust recommendation performance. The experimental results below also verify the effectiveness of this idea.
>
> To verify the robustness of EAPO to a decline in expert model quality, we conducted supplementary experiments (Figure 2(c) in the paper). We reduced the quality of the expert model by decreasing the training data (using 75% and 50% of the original training set). The experimental results are as follows:
>
> | Method | HR@1 | HR@5 | NDCG@5 |
> | :--- | :--- | :--- | :--- |
> | **EAPO (100% data)** | **0.5760** | **0.6977** | **0.6331** |
> | EAPO (w/ 75% data expert) | 0.5654 | 0.6937 | 0.6325 |
> | EAPO (w/ 50% data expert) | 0.5474 | 0.6723 | 0.6302 |
> | S-DPO| 0.4841 | 0.6597 | 0.5781 |
>
> As can be seen from the results in the table, even when the expert model training data is reduced to half the original size, the performance of EAPO only decreases slightly and is still significantly better than all other baseline methods. This phenomenon powerfully illustrates that EAPO is less sensitive to the quality of expert models in practical applications, demonstrating its strong robustness and generalization ability. As long as the expert model can provide relatively accurate guidance on preferences, EAPO can effectively benefit from it, thereby improving the overall recommendation performance.
>
> Finally, the example of "mobile vs. USB-C charging cable" mentioned by the reviewers is excellent, demonstrating the difference between semantic space and collaborative space, which precisely confirms EAPO's core contribution. LLMs might consider them trivially close in semantic space. However, in the actual collaborative filtering space in the recommendation domain, user preferences can be counterintuitive: for example, a user who buys an iPhone 15 (collaborative signal A) might not buy a USB-C charging cable (signal B, because the iPhone uses a Lightning interface), but instead would buy AirPods (signal C). For an LLM lacking this collaborative knowledge, it would incorrectly recommend B. EAPO's expert models (such as SASRec) precisely capture these massive collaborative co-occurrence patterns, providing LLMs with valuable domain preference knowledge that the LLM lacks. EAPO utilizes collaborative signals to correct the semantic bias of the LLM, which is not about introducing noise, but rather the key to achieving complementary advantages.

---

> ### Author Response · Authors · 2025-11-21
>
> ### **W4: Training Overhead**
>
> Thank you very much for your concern about model efficiency. To analyze the potential additional overhead of the expert model we introduced, we quantified the overhead introduced by the expert model during the training phase after the large model:
>
> | Training Configuration | Average Training Time per Step (s/iter) | Additional Overhead | Relative Time Impact |
> | :--- | :--- | :--- | :--- |
> | Baseline Model (No Expert) | 21.89s | - | - |
> | **EAPO (Introducing Expert Model)** | **21.90s** | **\~10.1ms** | **\~0.046%** |
>
> As shown in the table, the time overhead introduced by the lightweight expert model (approximately 10MB in size) during training is only 0.046%, which is almost negligible and adds virtually no additional GPU memory burden. Furthermore, even after removing these negligible overheads, the performance is improved by 19%, demonstrating its value in practical applications.
>
> | Methods | HR@1 | HR@5 | NDCG@5 |
> | :--- | :--- | :--- | :--- |
> | **EAPO** | **0.5760** | **0.6977** | **0.6331** |
> | S-DPO | 0.4841 | 0.6597 | 0.5781 |
>
> Furthermore, as described in the paper, we employ a lightweight expert model (approximately 10MB). With the current dataset size, pre-training a SASRec expert model takes approximately 4.6 minutes. Compared to the training cost and overhead of larger models, this is a one-time, almost negligible cost. Therefore, we believe this trade-off is extremely valuable and practically significant.
>
> ### **W5: $\gamma$ Selection Strategy**
> Thank you for your question regarding this key parameter. We want to clarify first that $\gamma$ is not a traditional hyperparameter like $U_{\beta}$, but rather a theoretical anchor point with a clearly defined setting method based on our Sec 3.1 theoretical analysis.
>
> Our core theoretical finding (Sec 3.1) is that $\beta$ has a non-monotonic relationship with the growth (i.e., gradient) of the reward difference $\Delta r$. There exists a critical point $\beta_c = z_c / \Delta r \approx 1.278 / \Delta r$.
>
>   * When $\beta < \beta_c$, increasing $\beta$ accelerates the growth of $\Delta r$.
>   * When $\beta > \beta_c$, increasing $\beta$ actually slows down the growth of $\Delta r$.
>
> The core idea of ​​EAPO's adaptive strategy is as follows: for difficult samples with small differences ($\Delta r$), we want $\beta_c$ to be larger, so we use a smaller $\beta$ to accelerate learning; for easy samples with large differences ($\Delta r$), we want $\beta_c$ to be smaller, so we use a larger $\beta$ to slow down learning (preventing overfitting). For this mechanism to work stably, we need to ensure that all our $\beta$ values ​​fall within the monotonically decreasing interval $\beta > \beta_c$.
>
> As described in Sec 3.1 of the paper, in the later stages of training, the difference in reward $\Delta r$ between positive and negative samples tends to stabilize and usually stably exceeds a certain constant value (for example, it stably exceeds 5 in our chosen dataset). $\gamma$ is precisely the constant value that this model stably exceeds during training. The process of choosing $\gamma$ is intuitive and principled: it can be determined simply by observing the distribution of $\Delta r$ in the pre-trained model. In our experiments (as shown in the ablation study table below), the value of $\gamma$ directly determines the lower bound of the adaptive $\beta$ ($\beta_{min} = z_c / \gamma$). A larger $\gamma$ allows $\beta$ to cover a wider dynamic range, enabling the model to handle samples of varying difficulty more finely. Experiments confirm that as $\gamma$ approaches the true distribution boundary of the data, the model performance steadily improves.
>
> | Methods | HR@1 | HR@5 | NDCG@5 |
> | :--- | :--- | :--- | :--- |
> | EAPO ($\gamma=4$) | **0.5760** | **0.6977** | **0.6331** |
> | EAPO ($\gamma=3$) | 0.5682 | 0.6915 | 0.6254 |
> | EAPO ($\gamma=2$) | 0.5543 | 0.6802 | 0.6117 |

---

> ### Author Response · Authors · 2025-11-21
>
> ### **W6: Regarding Experimental Rigor**
> We thank the reviewers for their meticulous consideration of experimental rigor. Due to computational resource limitations, the experiments in the initial draft were based on the results of a single run (random seed 1234). However, we recognize the importance of multi-run seeded experiments. During the rebuttal, we supplemented the results with averages of EAPO and S-DPO run on three different random seeds (42, 1234, 2025) to demonstrate that the performance improvement of our method is stable and statistically significant. The results are shown below:
>
> | Method | HR@1 | HR@5 | NDCG@5 |
> | :--- | :--- | :- | :- |
> | **EAPO** | **0.5717** | **0.7089** | **0.6381** |
> | S-DPO | 0.4952 | 0.6487 | 0.5804 |
>
> The results show that the overall performance is stable, demonstrating the robustness and consistent performance advantage of our method.
>
> The two LLM baselines with poor performance in Table 2 of the paper are the Zero-shot baselines. We have explicitly stated this in Appendix B.1.2. **We included these baselines to establish the lower bound** of performance for general LLMs that are not task-adapted. EAPO's primary comparison targets are not these zero-shot models, but rather state-of-the-art methods that have also undergone SFT and preference optimization. Because the untuned 8B model performs poorly, we did not perform much additional processing. Our significant 22.58% improvement is achieved by comparing EAPO (HR@1: 0.5760) with S-DPO (HR@1: 0.4841), a fair and robust comparison. To demonstrate the performance of the LLAMA3-8B model after few-shot training, we conducted the following experiments. It can be seen that while few-shot training can improve performance to some extent, the limitations of the base model limit its effectiveness.
>
> | Method | HR@1 | HR@5 | NDCG@5 |
> | :-| :- | :- | :- |
> | **EAPO** | **0.5760** | **0.6977** | **0.6331** |
> | Zero Shot | 0.0581 | 0.1795 | 0.1326 |
> | Few Shot | 0.0776 | 0.2067 | 0.1604 |
>
> In Appendix B.2.2, we describe our Prompting strategy, which employs a multi-template sampling strategy. For each sample, different expression formats are randomly selected from a library of 10 instruction templates for training. This approach follows the settings of S-DPO and LLaRa. To further improve transparency and reproducibility, we will include examples of the specific Prompt templates we used in the appendix of the final draft.
>
> ### **W7: Larger Parameter Models**
> Although limited by computational resources, we have not yet performed full fine-tuning on the 30B+ model; however, we have strong reasons to believe that the gains from EAPO will still hold true on larger-scale models. Regardless of how many parameters the **base LLM** increases (8B or 70B), they lack the **collaborative signals** specific to the dataset (i.e., the statistical regularity that user A usually buys Y after buying X). EAPO's core contribution is injecting this domain-specific collaborative knowledge into the LLM. This knowledge is not present in general pre-trained models; therefore, increasing the model size cannot replace the role of EAPO, and may even lead to a qualitative leap in performance due to scaling improvements. We will conduct large-scale model validation when suitable computational resources become available. To demonstrate the universality of EAPO to some extent, we validated its effectiveness on various optimization objectives, including IPO, CPO, and S-DPO (as shown in the table below). This indicates that EAPO is a general optimization paradigm that does not depend on the capabilities of a specific base model.
>
> | Metric | S-DPO | EA-SDPO | CPO | EA-CPO | IPO | EA-IPO |
> |---|---|---|---|---|---|---|
> | HR@1 | 0.4841 | **0.5558** (+14.81%) | 0.4341 | **0.4422** (+1.87%) | 0.4071 | **0.4209** (+3.39%) |
> | HR@5 | 0.6597 | **0.6829** (+3.52%) | 0.6345 | **0.6628** (+4.46%) | 0.7024 | **0.7143** (+1.69%) |
> | NDCG@5 | 0.5781 | **0.6118** (+5.83%) | 0.5091 | **0.5359** (+5.26%) | 0.5456 | **0.5568** (+2.05%) |
>
> Finally, to address your concerns, we expanded the model scope during Rebuttal by introducing the Qwen 2.5-7B model, which has a different architecture and excellent language capabilities, as a new base. Comparative experiments were conducted on the core datasets (Movies and TV).
>
> The experimental results are shown in the table below:
>
> | Base Model | Method | HR@1 | HR@5 | NDCG@5 |
> | :- | :- | :- | :- | :- |
> | **Llama-3-8B** | S-DPO | 0.4841 | 0.6597 | 0.5781 |
> | | **EAPO** | **0.5760** | **0.6977** | **0.6331** |
> | **Qwen 2.5-7B** | S-DPO | 0.5056 | 0.6780 | 0.6012 |
> | | **EAPO** | **0.5835** | **0.7240** | **0.6645** |
>
> The results show that EAPO provides performance comparable to Llama-3 on Qwen 2.5. The consistent performance leap (approximately 15.4% improvement in HR@1) validates the effectiveness of EAPO as a general optimization paradigm. This demonstrates that our approach is independent of specific base model capabilities or architectures.

---

> > ### Comment · Reviewer_mcur · 2025-11-28
> >
> > I thank the author for explanations and additional experiments. These will make the paper stronger.
> >
> > Among other things, for the training question, there has been a miscommunication, I know that therec model is small, but when I say training complexity increases, I mean training an extra model is non trivial because one would need to then also make sure those models are trained properly and goes to convergence etc. Disentangling the effect of an extra model to be trained inside the process is not always simple. Precisely why folks tend to use grpo type algorithms (without the value model) or dpo type algorithms (without value and reward models)  instead of rlhf lately. I was not refering to computational complexity of training. However I do not buy the authors argument that model size has no effect on the performance improvement delta of an algorithm. Authors have a strawman argument that `This knowledge is not present in general pre-trained models` . Yes that is perhaps possible if the the items are extremely specific, but not necessarily true for more general case and therefore training a similarly sized model with no immediate difference in expressive power wont prove much. However, I did not ask for extra experiments for this and I appreciate authors attempt with the qwen model. Small nit, iphone 15 _does_ use a usb c (but I see your point). My original issue was that the expert can introduce noise (noise could be unintended ranking signals from the rec model), if it is not of high quality, reducing the quality of the model by reducing the number of data points is a strange proposition, we dont know anything about the new model. Do we know any particular fine grained behavior has changed? Maybe the recommendation dataset was not complex, and it did not degrade the rec model enough.
> >
> > Nevertheless, overall I have increased my score.

---

### Author Response · Authors · 2025-12-01

We sincerely thank you for your careful review and the time you spent evaluating this work. We are also very grateful to the reviewer for clearly indicating that the main concerns were addressed. To summarize our responses and updates to the paper, we have summarized our responses below:

**Reviewer mcur:**

* Reviewer mcur raised concerns about the need for a stronger ranking baseline (such as LiPO). To address this, we supplemented our comparative experiments with LiPO, showing that EAPO achieved superior performance (HR@1 0.5760 vs 0.5483). We clarified the fundamental difference between the two: LiPO focuses on optimizing ranking within the semantic space, while EAPO explicitly injects the collaborative signals missing in expert models.

* Reviewer mcur questioned the possibility that this method might lead to a degradation in generative capabilities (log probs). We performed a thorough log shift analysis, demonstrating that EAPO significantly widens the reward gap by penalizing rejected items while maintaining the stability of the chosen item probability (EAPO average log probability -14.1, DPO -13.6), ensuring that the generation quality is not compromised.

* Reviewer mcur questioned training complexity and cost. We quantified the training by showing that the lightweight expert model only adds approximately 0.046% to the single-step time cost, and the pre-training time is negligible (approximately 4.6 minutes).

**Reviewer gnAK:**

* Reviewer gnAK questioned the novelty compared to $\beta$-DPO and MallowsPO. We elucidate the fundamental difference: EAPO uses stable external expert signals (cooperative knowledge) instead of the unstable internal self-referential signals used in $\beta$-DPO. We added ablation experiments demonstrating that this expert-guided instance-level strategy significantly outperforms batch-level and heuristic dynamic $\beta$ methods.

* Reviewer gnAK focuses on robustness to expert model quality. We conducted sensitivity analyses using experts trained with different proportions of data (50% and 75%), demonstrating that EAPO maintains high performance even when the expert model is not optimal, confirming its robustness.

**Reviewer E7tY:**

* Reviewer E7tY suggested experiments on different backbone models. We extended the experiments to Qwen 2.5-7B (except Llama-3), observing consistent performance improvements (15.4% improvement in HR@1), validating EAPO's cross-model generalization ability.

* Reviewer E7tY questioned the choice between instance-level and batch-level adaptation. We provide an additional theoretical ground based on the critical point $\beta_c$ and add comparative experiments showing that instance-level adaptation captures fine-grained preference structures better than batch-level strategies.

* Reviewer E7tY inquired about the implementation details of the S-DPO setup. We clarified the choice of the number of negative samples ($K=3$) and the margin calculation method, and corrected citation formatting issues in the revised manuscript.

**Reviewer CkM8:**

* Reviewer CkM8 inquired about sensitivity to expert model architectures. We conducted additional experiments using different expert backbone networks (DeepMF, SASRec, CL4Rec). The results show that DeepMF (pure collaborative filtering) performs best, confirming that the injection of collaborative signals is the core driver of performance improvement.

* Reviewer CkM8 questioned generalization beyond recommendation tasks. We applied the EAPO framework to a text summarization task (TL;DR dataset), using a reward model as the expert. Results show that EAPO achieved a 54.5% win rate compared to DPO, demonstrating its potential as a general optimization paradigm.

* Reviewer CkM8 questioned the assumption of a fixed reward gradient. We clarify in Appendix A.3 that this is a first-order Taylor expansion used to determine the *direction* of the $\beta$ adjustment, ensuring that the theoretical conclusions regarding the critical point $\beta_c$ remain robust.

We believe we have thoroughly addressed all the major concerns raised by the reviewers and made substantial updates to the manuscript (including new experiments on Qwen 2.5, the summarization task, and different expert architectures), which will be supplemented in the final version. We sincerely thank the reviewers for their insightful feedback, which significantly improved the technical rigor and breadth of our work.

* **Special Statement:**
In light of the OpenReview system malfunction that compromised the double-blindreview process, we solemnly declare:

     We have strictly adhered to the code of conduct regarding the fairness and integrity of the review process and did not exploit this vulnerability to ascertain any reviewers. **The discussion and subsequent score improvement (Reviewer mcur indicated an intention to increase the score) exclusively resulted from interactions through the official public channels on OpenReview.**

---

### Meta-Review · Area_Chair_UNii · 2026-01-05

**Summary:**

This paper studies expert-guided adaptive preference optimization for recommender systems. The goal is to improve preference alignment and recommendation performance by introducing a dynamic optimization mechanism inspired by recent work on LLM preference learning. Reviewers generally agree that:

- The problem of preference optimization for recommender systems is important and relevant in practice.
- The paper presents a clear framework and includes empirical evaluations on recommendation benchmarks.
- The experimental results show improvements over several baselines, suggesting that the method can work in practice.
- Most presentation-related issues are addressed in the rebuttal.

However, a critical concern raised by **Reviewer gnAK and Reviewer CkM8** remains unresolved after the rebuttal and discussion. Both reviewers point out that the central idea of using dynamic beta to capture preference diversity has already been explored in prior work on general LLM preference learning. From this perspective, they argue that the paper does not provide enough references or direct comparisons to clearly show how it differs from existing methods. As a result, applying similar ideas to recommender systems appears incremental.

Although the authors attempt to explain the differences in the rebuttal, the response does not clearly show how the proposed method goes beyond existing approaches. Therefore, the novelty concerns raised by **Reviewer gnAK and Reviewer CkM8** are not fully addressed and remain central issues in the final evaluation.

**Reviewer Concerns:**

- **Reviewer gnAK** raises major concerns about the novelty of the paper. In particular, the reviewer points out that the key idea of using dynamic beta to capture preference diversity has already been explored in prior work on general LLM preference learning (e.g., dynamic or heuristic adaptations of preference strength). The reviewer argues that the paper lacks sufficient references and direct comparisons to these existing methods, and that applying similar ideas to recommender systems appears incremental. The authors respond by clarifying the motivation and implementation details, but the rebuttal does not clearly demonstrate how the proposed approach differs from or improves upon prior work, leaving this concern unresolved.
- **Reviewer mcur** finds the problem important and the results promising, but raises concerns about baselines and objective clarity, which are only partially addressed in the rebuttal.
- **Reviewer EZTy** is generally positive, with minor concerns about experiments and presentation that are addressed.
- **Reviewer CkM8** is positive about the empirical performance but questions the novelty of some components, and this concern is not fully addressed.

Overall, while the rebuttal resolves several issues, it does not fully address some key concerns raised by the reviewers.

**Reviewer Scores:**

- I would expect **Reviewer gnAK** and  **Reviewer CkM8** to maintain a negative score due to unresolved concerns about novelty and contribution.
-  I would expect **Reviewer mcur** and **Reviewer EZTy** to maintain a slightly positive score after the rebuttal.

Above all, I think this is a borderline paper. But concerning the unresolved issues mentioned above. I have to recommend rejecting this paper.

---

### Decision · Program_Chairs · 2026-01-26

Reject